# DreamDojo: A Generalist Robot World Model from Large-Scale Human Videos

Shenyuan Gao [* 1 2]  William Liang [* 1 3]  Kaiyuan Zheng [1 4]  Ayaan Malik [1 5]  Seonghyeon Ye [1 6]  Sihyun Yu [6]
Wei-Cheng Tseng [1 7]  Yuzhu Dong [1]  Kaichun Mo [1]  Chen-Hsuan Lin [1]  Jiannan Xiang [8]  Yuqi Xie [1]
Ruijie Zheng [1]  Dantong Niu [1 3]  Pooya Jannaty [1]  Jinwei Gu [1]  Jun Zhang [2]  Jitendra Malik [3]  Pieter Abbeel [3]
Ming-Yu Liu [1]  Yuke Zhu [1 9]  Joel Jang [1]  Linxi Fan [1]

dreamdojo-world.github.io

## Abstract

Being able to simulate the outcomes of actions in varied environments will revolutionize the development of generalist agents at scale. However, modeling these world dynamics, especially for dexterous robotics tasks, poses significant challenges due to limited data coverage and scarce action labels. As an endeavor towards this end, we introduce DreamDojo, a generalist world model that learns diverse interactions and dexterous controls from 44k hours of egocentric human videos. Our data mixture represents the largest video dataset to date for world model pretraining, spanning a wide range of daily scenarios with diverse objects and skills. To address the scarcity of action labels, we introduce continuous latent actions as unified proxy actions, enhancing interaction knowledge transfer from unlabeled videos. After post-training on small-scale target robot data, DreamDojo demonstrates a strong understanding of physics and precise action controllability. We also devise a distillation pipeline that accelerates DreamDojo to a real-time speed of 10.93 FPS and further improves consistency to the context. Our work enables several important applications based on generative world models, including live teleoperation, policy evaluation, and model-based planning. Systematic evaluation on multiple challenging out-of-distribution (OOD) benchmarks verifies the significance of our method for simulating open-world, contact-rich tasks, paving the way for general-purpose robot world models.

## 1. Introduction

World models, which predict futures based on actions, have emerged as a pivotal component in the development of generalist robots (Sutton, 1991; LeCun, 2022; Hu et al., 2023b; Richens et al., 2025). Recent advances in video generation (Ali et al., 2025; Wan et al., 2025) have ushered in a new era of video world models, in which future states are represented as video frames (Parker-Holder & Fruchter, 2025; Russell et al., 2025; Sun et al., 2025b). However, current world models primarily plateau at discrete controls, while the high-dimensional action spaces for contact-rich robot tasks have yet to make similar progress. Unlike game and driving domains, robot data often has limited coverage due to hardware divergence and collection cost. The nearly infinite variety of real-world environments can easily exceed the distribution of the available robot data. Additionally, existing datasets predominantly consist of expert demonstrations, lacking the stochasticity in intentions necessary for learning strong action controllability. As a result, existing world models remain confined to simulating observed setups and are often unresponsive to counterfactual actions, constraining their applicability for diverse scenarios and complex tasks.

In this work, we introduce **DreamDojo**, a generalist world model for open-world dexterous robot tasks. Unlike previous methods that typically rely on teleoperation data, we exploit *human videos* for pretraining. Despite the embodiment gap, the underlying physics during interactions is largely consistent between humans and robots, enabling effective knowledge transfer. Therefore, we curate the *largest* egocentric human video dataset to date, **DreamDojo-HV** (Human Videos), which comprises 44k hours of video sequences, surpassing the datasets used in prior works by several orders of magnitude. In addition to its scale, DreamDojo-HV incorporates an exceptionally diverse range of activities, encompassing approximately $96\times$ more skills and $2,000\times$ more scenes than the most diverse public datasets for robot learning (Khazatsky et al., 2024; Bu et al., 2025a). This provides us with a rich corpus for learning physics and

---
*Equal contribution  [1]NVIDIA  [2]HKUST  [3]UC Berkeley  [4]University of Washington  [5]Stanford University  [6]KAIST  [7]University of Toronto  [8]UC San Diego  [9]UT Austin. Correspondence to: Shenyuan Gao <sygao@connect.ust.hk>.

*Proceedings of the 43rd International Conference on Machine Learning*, Seoul, South Korea. PMLR 306, 2026. Copyright 2026 by the author(s).

dynamics about different kinds of interactions.

Nevertheless, fine-grained action labels are much scarcer than raw videos when scaling up. Naively training on passive videos overlooks the causality between video observations and actions, leading to inferior knowledge transfer for action-conditioned world simulation. Moreover, converting various action formats into a unified one entails inevitable engineering effort. To address these challenges, we introduce *continuous latent actions* (Gao et al., 2025) as unified proxy actions for all videos. The proposed latent action model can extract semantically meaningful actions between frames in a self-supervised manner, ensuring effective transfer of both physics and controllability as the data scales to the internet level. Through rigorous designs of model architecture and training recipe, DreamDojo is able to acquire a comprehensive understanding of physics, achieving plausible simulations across diverse environments and fine-grained controllability over continuous robot actions.

To achieve real-time prediction without visual degradation, we further introduce a distillation pipeline following the Self Forcing paradigm (Huang et al., 2025b). Our distillation also enhances the long-horizon consistency by efficiently modeling a short temporal context. The resulting model can autoregressively predict future frames at a resolution of $640 \times 480$ at 10.93 FPS for an arbitrary horizon, significantly reducing the cost for various downstream applications such as live teleoperation and model-based planning.

In summary, our main contributions include: (1) **A large-scale video dataset**, DreamDojo-HV, that accumulates 44k hours of egocentric experiences from a wide spectrum of daily activities. To the best of our knowledge, this is the *largest* and most diverse data corpus to date for world model learning. (2) **A generalist world model** for general-purpose robots. By scaling up human videos and introducing continuous latent actions as unified proxy, we present DreamDojo, the *first* world model of its kind that shows zero-shot generalization to unseen objects and novel environments. (3) **A distillation pipeline** that enables efficient autoregressive prediction and improves context consistency. The final model can be interacted with for more than 1 minute in real time without degradation. (4) **Multiple downstream applications** highlight the potential of DreamDojo in performing live teleoperation, policy evaluation, model-based planning, *etc*., accelerating the development of robot policies.

## 2. Preliminary

**Interactive world model.** The objective of an interactive world model is to infer future states based on actions. Formally, given an action $a \in \mathcal{A}$, the interactive world model acts as a state transition function that samples the next state:

$$s_{t+1} \sim p(\cdot | s_t, a_t), \qquad (1)$$

where $p : \mathcal{S} \times \mathcal{A} \to \Delta(\mathcal{S})$ is the transition distribution. In this paper, the term "world model" refers specifically to this category unless otherwise stated.

**Cosmos-Predict2.5 model.** We establish our world model on the pretrained Cosmos-Predict2.5 model (Ali et al., 2025), a latent video diffusion model that predicts future frames with text and conditional frame inputs. The Cosmos-Predict2.5 model operates in the continuous latent space produced by WAN2.1 tokenizer (Wan et al., 2025). It injects language and timestep conditions into each DiT block (Peebles & Xie, 2023). The text embedding is processed by cross-attention layers, while the timestep information is first encoded by sinusoidal embeddings, projected by a lightweight MLP, and then used by adaptive layer normalization for dynamic modulations (scale, shift, gate) (Ali et al., 2025). The whole network is trained using flow matching loss (Lipman et al., 2022) for its simplicity and effectiveness. Specifically, given the noise corrupted video latent $\mathbf{x}_t$ at timestep $t$, the flow matching loss minimizes the prediction error with the ground-truth velocity $\mathbf{v}_t$:

$$\mathcal{L}_{\text{flow}}(\theta) = \mathbb{E}_{\mathbf{x}, \epsilon, \mathbf{c}, t} \left\| \mathbf{u}(\mathbf{x}_t, t, \mathbf{c}; \theta) - \mathbf{v}_t \right\|^2, \qquad (2)$$

where $\mathbf{v}_t$ is a difference between the noise $\epsilon$ and the clean sample $\mathbf{x}$ (*i.e.*, $\mathbf{v}_t = \epsilon - \mathbf{x}$), $\mathbf{c}$ denotes any conditions (*e.g.*, text, conditional frames, and actions for world models), and $\mathbf{u}(\cdot; \theta)$ represents the denoiser parametrized by $\theta$.

## 3. Approach

### 3.1. Overview

In this section, we first introduce the features of our dataset in Section 3.2, and then describe the architecture of Dream-Dojo and its training recipe in Section 3.3. Our whole training procedure consists of three phases: (1) **Pretraining from human videos.** At this stage, we curate three egocentric human datasets for pretraining: In-lab, EgoDex, and DreamDojo-HV. Continuous latent actions are introduced as conditions for all videos. (2) **Post-training on target robots.** To adapt DreamDojo to different embodiments, we reset the action conditioning layer and learn the new action space through finetuning. The post-training can be conducted on a target robot dataset collected from limited scenarios. (3) **Distillation.** Once DreamDojo has learned the target action space, a distillation process can be applied to improve real-time interactivity and context consistency.

### 3.2. DreamDojo-HV Dataset

Existing robot world models are primarily limited to in-distribution settings and fall short in generalizing to unseen interactions with new objects (Team et al., 2025; Zhang et al., 2025a). In essence, this limitation arises because most datasets only cover a relatively narrow distribution with

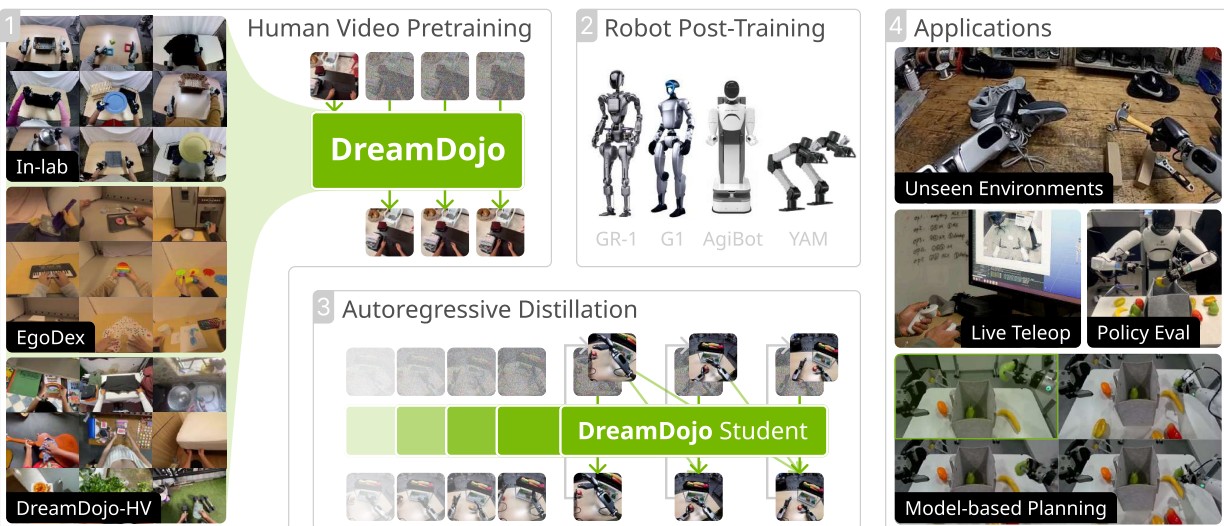

Figure 1. **DreamDojo overview.** DreamDojo acquires comprehensive physical knowledge from large-scale human datasets by utilizing latent actions as unified labels. After post-training and distillation on the target robots, our model can predict the future world in real time with continuous action controls. DreamDojo can robustly generalize to various objects and environments, facilitating large-scale policy evaluation without real-world deployment. It also enables live teleoperation and online model-based planning.

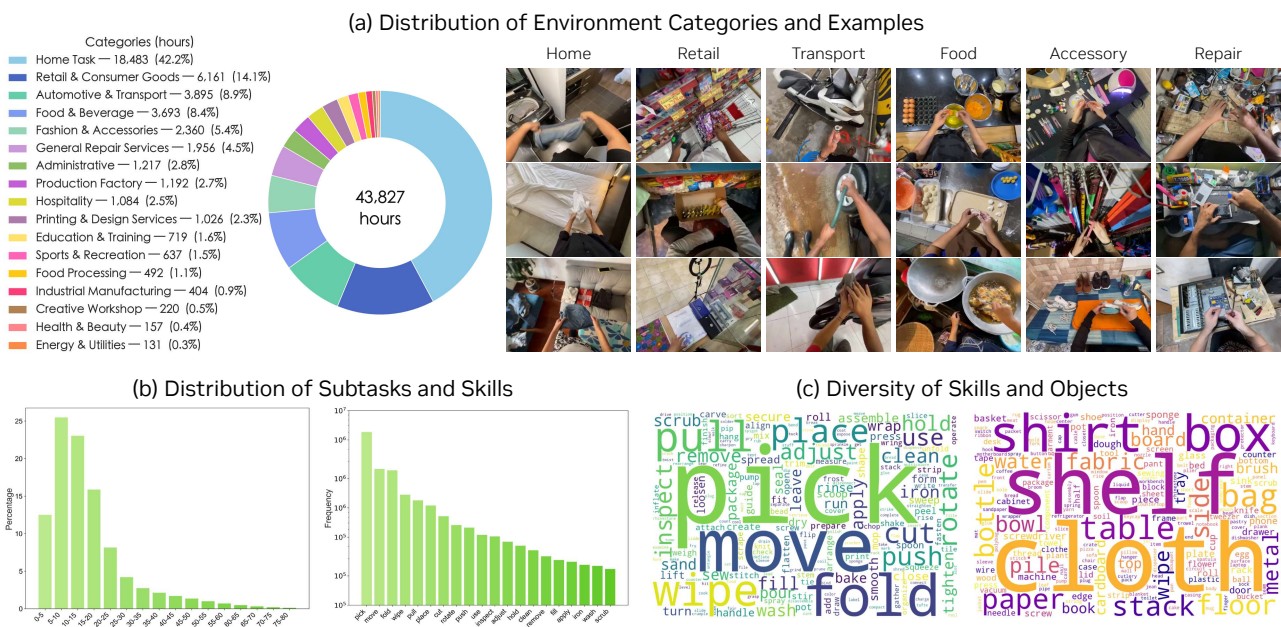

Figure 2. **Distribution analysis of DreamDojo-HV. (a)** Distribution of the scenarios and random examples from the most frequent categories. **(b)** [Left]: Distribution of subtask numbers within each video. Most videos involve long-horizon tasks that require multiple interactions to accomplish. [Right]: Representative skills in DreamDojo-HV and their frequencies. Our dataset covers a wide range of interaction types beyond pick-and-place. **(c)** Visualization of skill verbs and object names based on their frequency of occurrence in language annotations.

*Table 1.* **Scale and diversity comparison to existing large-scale datasets used by previous world models.** Our curated data mixture excels in both scale and diversity, encompassing 15× longer duration, 96× more skills, and 2,000× more scenes than the previously largest dataset for world model training. [†]Estimated by GPT based on the global language annotation of each video clip.

| Dataset | Type | Prior Works | # Hour | # Trajectory | # Skill | # Scene |
|---|---|---|---|---|---|---|
| RT-1 | Robot | IRASim, UniSim | 900 | 130k | 8 | 2 |
| BridgeData V2 | Robot | IRASim, UniSim, WorldGym, WMPE | 130 | 60.1k | 13 | 24 |
| Language-Table | Robot | IRASim, UniSim, HMA | 2.7k | 442k | - | - |
| RoboNet | Robot | iVideoGPT, IRASim | - | 162k | - | 10 |
| DROID | Robot | Ctrl-World, UWM | 350 | 76k | 86 | 564 |
| AgiBot-World | Robot | EnerVerse-AC | 2.9k | 1,000k | 87 | 106 |
| Nymeria | Human | PEVA, EgoTwin, EgoControl | 300 | 1.2k | - | 50 |
| In-lab | Human | - | 55 | 13.9k | 35 | 1 |
| EgoDex | Human | - | 829 | 30k | 194 | 5 |
| DreamDojo-HV | Human | - | 43,827 | 1,135k | 6,015[†] | 1,135k |
| **Our Mixture** | Human | - | 44,711 | 1,179k | ≥6,015[†] | ≥1,135k |

limited verbs, objects, and environments, thereby restricting the breadth of interaction patterns. As a result, training on these datasets often fails to preserve the model's abilities when extending to out-of-distribution scenarios.

To address this limitation, one might consider increasing the scale of real robot data. However, this may not be the most efficient approach to encompass all potential interaction types, as each new trajectory involves costly teleoperation. On the other hand, human videos, which can be captured during daily activities, emerge as a promising axis to empower robot learning (Zheng et al., 2025; Kareer et al., 2025; Ye et al., 2025b; Chen et al., 2025a) . Inspired by these studies, we scale up human videos as a pioneering step for world model pretraining. Our human data comes from three sources: (1) **In-lab** is collected in our tabletop settings at our laboratory to validate our core designs. The collectors are wearing Manus gloves with Vive Ultimate Tracker to capture precise hand poses, which can be readily retargeted to the GR-1 robot actions. It contains several new objects and new verbs that are unseen in our default robot training dataset. (2) **EgoDex** (Hoque et al., 2025) is a public dexterous human manipulation dataset with egocentric views recorded by Apple Vision Pro. It has 829 hours of egocentric videos with high-precision 3D hand and finger poses collected at the time of recording. It also contains a variety of everyday household objects. We include it to our data suite to enrich object variety. (3) **DreamDojo-HV** is a large-scale in-house dataset collected through crowdsourcing. It features a wide spectrum of loco-manipulation skills and extremely diverse environments, such as household, industrial, retail, educational, and administrative (see Figure 2 for the distribution), which significantly increases the scale and diversity of our data corpus. Each episode is annotated with a text that describes the task being performed.

As shown in Table 1, our final dataset comprises a total of 44,711 hours, making it the largest human interaction dataset to date for world model pretraining. It includes more

than 9,869 unique scenes, 6,015 unique tasks, and 43,237 unique objects, representing the majority of interactions in daily activities. Its scale and diversity also provide thorough coverage of various action distributions and increase the stochasticity of the future, thereby enhancing the controllability of actions. In Section 4.3, we demonstrate that increasing the data scale and diversity continues to enhance model performance across all benchmarks.

### 3.3. DreamDojo Generalist World Model

#### 3.3.1. MODEL ARCHITECTURE

Unlike conventional video generators (Wan et al., 2025; Ali et al., 2025), world models are distinct in their prioritization on action controllability (Yang et al., 2024b; 2025a; Huang et al., 2025a). Different from interactive games with discrete inputs (Parker-Holder et al., 2024), achieving genuine controllability for robot actions presents more challenges due to its high dimensionality and contact-rich nature.

To realize precise action following, we propose two improvements based on the original architecture. First, instead of using the absolute robot joint poses, we transform them into relative actions by rebaselining the inputs with the pose at the beginning of each latent frame (*i.e.*, every 4 timesteps). Since relative actions are often concentrated in a narrower space shared across various trajectories, this significantly reduces modeling complexity, thereby enhancing generalization to continuous and compositional robot actions.

Second, since the consequences of interactions strictly follow causality, observing future actions does not aid predictions at the current timestep but rather increases irrelevant noise. Therefore, instead of sending the entire relative action trajectory as a global condition for all latent frames, we inject actions into the latent frames as chunks (Huang et al., 2025a; Zhu et al., 2025b; Guo et al., 2025b). Specifically, since the temporal compression ratio of the WAN2.1 tokenizer (Wan et al., 2025) is 4 (*i.e.*, video latent $x^i$ cor-

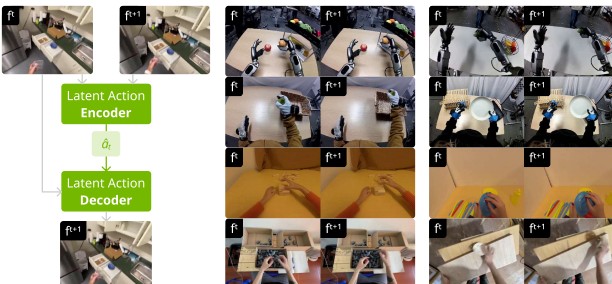

*Figure 3.* **Latent action model.** [Left]: The information bottleneck design of our latent action model enforces action disentanglement, producing a continuous latent vector that represents actions between frames. [Right]: We retrieve and group the frame pairs from different datasets that share the most similar latent actions. The embodiments are performing the same actions despite the significant differences in context.

responds to 4 frames $f^{i:i+4}$ in the pixel space, $x^{i+1}$ corresponds to $f^{i+4:i+8}$, and so on), we concatenate 4 consecutive actions $a^{t:t+4}$ as a chunk and send them to the corresponding latent frame together. This strong prior can greatly mitigate the causality confusion, thereby improving learning efficiency and ultimately enhancing controllability. We show the effects of these two designs above on both expert and counterfactual trajectories in Section D.8.

### 3.3.2. PRETRAINING FROM HUMAN VIDEOS

**Latent action as proxy action.** While our data suite includes comprehensive real-world activities, it does not come with fine-grained action annotations. To inherit the rich knowledge from this unlabeled dataset, one straightforward solution is to pretrain the model by passively predicting future frames. In our experiments, we found that this simple approach can transfer certain physical knowledge from human videos, resulting in more physically plausible modeling for unseen objects. Nevertheless, since world models must learn the consequences of actions, relying solely on actionless videos may lead to an inadequate understanding of the causality, ultimately resulting in inferior interactivity when adapting to the target robots.

To address the inefficient knowledge transfer caused by absence of action labels, it is crucial to derive pseudo labels from pixels that describe the current actions. While off-the-shelf models like HaMeR (Pavlakos et al., 2024) can extract hand poses at scale, these models struggle to represent actions beyond the hands (*e.g.*, arm movements and locomotions) and often face challenges in inferring hand positions in heavy occlusions. In addition, they primarily focus on low-level features of the human hands, which may hinder effective knowledge transfer to the target robot when there is a significant embodiment gap.

On the other hand, latent actions (Bruce et al., 2024; Ye et al., 2025b; Gao et al., 2025), which extract action in-

formation purely from videos in a self-supervised manner and provide consistent action interpretation across embodiments, have gained increasing attention recently. Inspired by (Gao et al., 2025), we adopt continuous latent actions due to their superiority in cross-embodiment generalization and efficient adaptability. We establish a latent action model as a VAE (Kingma & Welling, 2013) using the spatiotemporal Transformer architecture (Bruce et al., 2024). It has an information bottleneck design that can automatically disentangle the most critical action information from the context. Specifically, unlike the standard VAE, our VAE encoder takes two consecutive frames $f^{t:t+1}$, extracts spatiotemporal features, and projects the global features to a low-dimensional embedding $\hat{a}_t$. The VAE decoder receives this embedding along with the former frame $f^t$, aggregates the information and predicts the subsequent frame $f^{t+1}$. The entire VAE is supervised by the reconstruction loss of the later frame and the KL divergence. The compactness of the embedding and the regularization term together create an information bottleneck. To reconstruct the later frame, the model has to disentangle and compress the most critical motions between frames.

$$
\begin{aligned}
\mathcal{L}^{pred}_{\theta,\phi}(f^{t+1}) = \; & \mathbb{E}_{q_\phi(\hat{a}|f^{t:t+1})} \left[ \log p_\theta(f^{t+1}|\hat{a}, f^t) \right] \\
& - \beta \, D_{KL}(q_\phi(\hat{a}|f^{t:t+1}) || p(\hat{a})).
\end{aligned}
\tag{3}
$$

In egocentric human videos, we found that this embedding particularly captures the human actions and can be transferred across embodiments (see Figure 3). Ultimately, we are able to utilize latent actions as unified proxy labels for world models. During pretraining, we condition each latent frame on the chunked latent actions. Concretely, after extracting latent actions from consecutive frames, we project them using a lightweight MLP to match the dimensions of the timestep embeddings. The last layer of the action MLP is initialized with zeros to avoid perturbing the pretrained model state at the beginning of training (Zhang et al., 2023), which we empirically found leads to improved physics. The projected embeddings are added with the timestep embeddings and then fed into the adaptive layer normalization within each DiT block.

**Training objective.** As mentioned in Section 2, Cosmos-Predict2.5 employs the standard flow matching loss (Equation (2)) as its training objective. However, this individual supervision of each frame overlooks the temporal correlation between video frames, which could provide a more direct signal for learning object dynamics and action following. Inspired by previous studies (Wang et al., 2024b; Gao et al., 2024; Yang et al., 2025a), we modify Equation (2) to a new loss that aims to match the ground-truth temporal transitions. Let $\mathbf{z}_t = \mathbf{u}(\mathbf{x}_t, t, \mathbf{c}; \theta)$ represent the predicted velocity, then the proposed temporal consistency loss can

be expressed as:

$$\mathcal{L}_{\text{temporal}}(\theta) = \mathbb{E}\Big[ \sum_{i=1}^{K-1} \big\| (z^{i+1} - z^i) - (v^{i+1} - v^i)) \big\|^2 \Big].$$
(4)

Here, $K$ is the total length of the video latent, $[z^1, z^2, \ldots, z^k]$ and $[v^1, v^2, \ldots, v^k]$ are frames in $\mathbf{z}_t$ and $\mathbf{v}_t$, respectively. In practice, we found that this loss term not only accelerates the learning of action controllability but also effectively enhances object completeness and reduces artifacts. Therefore, our final training objective becomes:

$$\mathcal{L}_{\text{final}}(\theta) = \mathcal{L}_{\text{flow}}(\theta) + \lambda\,\mathcal{L}_{\text{temporal}}(\theta),$$
(5)

where $\lambda > 0$ is a trade-off coefficient to balance the optimization. We use $\lambda = 0.1$ in our experiments.

### 3.3.3. POST-TRAINING ON TARGET ROBOTS

While learning from human videos exposes the model to a wide range of physics interactions, we still require a post-training stage on the target robot data to adapt our model for downstream applications. To achieve this, we flatten the ground-truth actions of the target robot into a sequence and project the entire sequence through the action MLP. To match the target action space, we reinitialize the first layer of the action MLP and fully finetune it along with all other pretrained weights. Thanks to strong pretraining, the target robot dataset can be collected in limited domains at a small scale while still achieving zero-shot generalization after finetuning. The continuity of our latent action space also ensures better adaptation results compared to other variants (Gao et al., 2025).

### 3.3.4. DISTILLATION

In order to unlock capabilities like live teleoperation and online model-based planning, our world model must run autoregressively in real time (Huang, 2025). However, existing video diffusion models are often limited in achieving this due to (1) their bidirectional attention architecture, which defines a fixed horizon length, and (2) a large number of denoising timesteps (*e.g.*, 50), which severely hampers inference speed.

Thus, we introduce an additional distillation stage that converts the foundation DreamDojo model into an autoregressive, few-step diffusion model. We build on the process introduced by Self Forcing (Huang et al., 2025b), which uses two training stages to distill teacher $G_{\text{teacher}}$ to student $G_{\text{student}}$. We construct $G_{\text{student}}$ with the same architecture and model weights of $G_{\text{teacher}}$, with the exception of the bidirectional attention mechanism, which is replaced with causal attention, and the timestep schedule, which is shortened to a few steps (*e.g.*, 4). In the first "warmup" stage,

we regress student predictions to match ODE solutions generated by our teacher and train via teacher forcing, *i.e.*, its context consists of latents generated by the teacher. Afterwards, in the second "distillation" stage, we construct the student context with its own previously-generated latents, and supervise with a distribution matching loss (Yin et al., 2024b) based on the KL divergence between real (teacher) and fake (student) distributions. To compute its gradient, we use real and fake models $s_{\text{real}}$ and $s_{\text{fake}}$,

$$\nabla\mathcal{L}_{\text{distill}} = -\mathbb{E}_{z,t}\left[ (s_{\text{real}}(x_t, t) - s_{\text{fake}}(x_t, t)) \frac{dG_{\text{student}}}{d\theta} \right],$$
(6)

where $z \sim \mathcal{N}(0, I)$ is noise, $x_t$ is produced via forward diffusion with $G_{\text{student}}$ starting at $z$, and $s_{\text{real}}$ is estimated by $G_{\text{teacher}}$ whereas $s_{\text{fake}}$ is estimated by a model trained on the predictions of $G_{\text{student}}$.

This process minimizes the train-test distribution mismatch, as the student is trained to generate from its previous outputs. To mitigate degradation over long horizons, we also propose to have the student generate more frames than the horizon of the teacher, simulating longer student rollouts and thus further minimizing the train-test discrepancy.

## 4. Experiments

In this section, we conduct extensive experiments to demonstrate DreamDojo's strengths. Specifically, we aim to answer the following questions: (1) Compared to actionless pretraining, can latent actions enable more effective transfer from human videos? (Section 4.2) (2) Can more diverse data help generalize to new types of physical interaction and scenarios? (Section 4.3 and Section D.6) (3) Can our architectural design and training objective improve the action-conditioned prediction? (Section D.8) (4) Can our distillation pipeline accelerate and stabilize long-horizon interactions? (Section 4.4) (5) Can DreamDojo empower downstream applications, including policy evaluation, test-time search, and live teleoperation? (Section 4.5)

### 4.1. Benchmark Construction

To demonstrate the effectiveness of our method, we conduct the first systematic evaluation that emphasizes out-of-distribution scenarios and counterfactual actions. To be specific, we mirror the diverse and novel interactions in the three human datasets and construct three corresponding evaluation sets using the Fourier GR-1 humanoid robot: (1) **In-lab Eval**, (2) **EgoDex Eval**, (3) **DreamDojo-HV Eval**. We make every effort to replicate the objects observed in the human videos, allowing our robots to perform the same interactions that reflect similar underlying physics. We also collect a (4) **Counterfactual Eval** set that focuses on counterfactual actions not present in current robot learn-

ing datasets, such as patting a toy or reaching toward an object but missing it. To assess DreamDojo's generalization to diverse environmental changes, we further employ Gemini 2.5 Flash Image (*a.k.a.* Nano Banana) (Comanici et al., 2025) to edit the backgrounds of EgoDex Eval and DreamDojo-HV Eval to replicate typical observations in the original datasets. This results in (5) **EgoDex-novel Eval** and (6) **DreamDojo-HV-novel Eval**, with each consisting of 25 samples. A glimpse of the samples from our benchmarks is provided in Figure 7.

## 4.2. Effects of Different Action Conditions

We conduct experiments on both In-lab Eval and EgoDex Eval. To demonstrate the efficacy of the proposed latent action conditioning, we compare our method with three representative baselines: (1) **Without pretraining.** In this setup, the model is initialized from Cosmos-Predict2.5 directly for post-training without observing the human videos. (2) **Action-free pretraining.** In this baseline, we pretrain the world model on unlabeled videos as passive future prediction. The pretrained model is then used for post-training. (3) **Ground-truth action conditioning.** In this baseline, we pretrain the world model with ground-truth action conditioning. This is an ideal setting, where additional equipment is required to obtain high-quality action labels. Specifically, on the In-lab dataset, we condition our model on retargeted GR-1 actions captured by Manus gloves with a Vive Ultimate Tracker, which are mapped to the real GR-1 action specifications for each degree of freedom. The EgoDex dataset utilizes Apple Vision Pro to capture hand poses, which are subsequently transformed into MANO (Romero et al., 2022) by ourselves as conditions during pretraining.

All compared methods are pretrained for 50k steps on the corresponding human datasets and post-trained for 25k steps on the in-distribution GR-1 dataset. The post-trained models are evaluated on In-lab Eval and EgoDex Eval, which contain novel objects and actions not present in the GR-1 training dataset. The results are reported in Table 3. While pretraining on human videos through action-free video prediction brings marginal benefits, introducing latent actions significantly narrows the gap to the ideal scenario where ground-truth action labels are available. We also provide the PSNR curves throughout the post-training stage in the Appendix. Pretraining with latent actions can reach a much higher upper bound than action-free pretraining and without pretraining. Note that, though MANO actions can also be extracted from videos (Pavlakos et al., 2024), it is likely not as precise as using the Apple Vision Pro to derive the MANO labels in Table 3. Hence, we choose latent actions as unified proxy actions for all human videos during pretraining.

## 4.3. Effects of Different Data Mixtures

We also conduct a dataset ablation to validate the benefits of increasing data diversity. Specifically, we pretrain our model on different data combinations for 50k steps, and then post-train on the GR-1 dataset for 25k steps. Unlike our final models, the sampling ratio is uniform across each dataset for the model variants in this ablation study. We evaluate the post-trained models on In-lab Eval, EgoDex Eval, and DreamDojo-HV Eval, which contain unseen object interactions, as well as on Counterfactual Eval, which features counterfactual actions. From the results in Table 2, increasing data diversity not only improves physics modeling but also enhances the controllability for out-of-distribution actions. We also conduct human preference study for unseen scenarios. See the Appendix for evaluation results.

## 4.4. Benefits of Distillation

To unlock real-time inference, we distill the GR-1 post-trained variant of DreamDojo-2B using the same GR-1 dataset. Stress testing the capabilities of this distillation process, we run both the teacher and student models on GR-1 Long Eval, generating 600 frames (1 minute) of long-horizon, multi-stage tasks. As seen in Table 4, our student model, despite being few-step and autoregressive, achieves performance close to that of the teacher while running at nearly $4\times$ faster when benchmarked on H100 GPUs. In addition, the autoregressive architecture of our student offers two extra advantages. First, since the student generates each latent frame autoregressively, it enables real-time streaming, which we demonstrate in Section 4.5 is crucial for multiple downstream applications. Second, different from the teacher that is conditioned on a single initial frame, the distilled model can naturally incorporate multiple frames as context, resulting in superior robustness to occlusions and camera shifts. See our Appendix for qualitative samples and the analysis of the generalization ability after distillation.

## 4.5. Downstream Applications

**Policy evaluation.** One of the most straightforward application of world models is for policy evaluation (Li et al., 2025d; Quevedo et al., 2025; Zbinden et al., 2025; Tseng et al., 2025; Team et al., 2025). In this work, we choose AgiBot fruit packing as a typical long-horizon task to verify whether DreamDojo can perform policy evaluation accurately. We train a single-view state-free variant of GR00T N1.5 (Bjorck et al., 2025) on the fruit packing dataset. We also post-train DreamDojo-2B on the same AgiBot dataset. Afterwards, we deploy different checkpoints from training to collect closed-loop rollouts in the real world. More implementation details are provided in the Appendix.

Following WorldEval (Li et al., 2025d) and SIMPLER (Li

*Table 2.* **Effects of using different data mixture.** Adding more human datasets to pretraining consistently improves the performance for both out-of-distribution scenarios and counterfactual actions, highlighting the the potential of our approach.

| Pretrained Model | In-lab Eval | | | EgoDex Eval | | | DreamDojo-HV Eval | | | Counterfactual Eval | | |
|---|---|---|---|---|---|---|---|---|---|---|---|---|
| | PSNR↑ | SSIM↑ | LPIPS↓ | PSNR↑ | SSIM↑ | LPIPS↓ | PSNR↑ | SSIM↑ | LPIPS↓ | PSNR↑ | SSIM↑ | LPIPS↓ |
| Cosmos-Predict2.5 | 20.576 | 0.774 | 0.222 | 19.952 | 0.787 | 0.219 | 18.274 | 0.754 | 0.236 | 20.472 | 0.802 | 0.190 |
| Data Mixture | | | | | | | | | | | | |
|   In-lab | 20.913 | 0.776 | 0.219 | 20.267 | 0.785 | 0.218 | 18.621 | 0.754 | 0.233 | 20.755 | 0.796 | 0.187 |
|   In-lab+EgoDex | 20.972 | 0.778 | 0.216 | 20.334 | 0.791 | 0.215 | 18.706 | 0.762 | 0.230 | 20.797 | 0.796 | 0.188 |
|   In-lab+EgoDex+DreamDojo-HV | 21.016 | 0.781 | 0.215 | 20.414 | 0.790 | 0.216 | 18.724 | 0.759 | 0.232 | 20.852 | 0.799 | 0.188 |
| DreamDojo-2B | 21.114 | 0.774 | 0.222 | 20.411 | 0.775 | 0.226 | 18.813 | 0.747 | 0.238 | 20.907 | 0.787 | 0.192 |
| DreamDojo-14B | 21.413 | 0.788 | 0.208 | 20.525 | 0.787 | 0.213 | 18.924 | 0.751 | 0.228 | 21.087 | 0.793 | 0.185 |

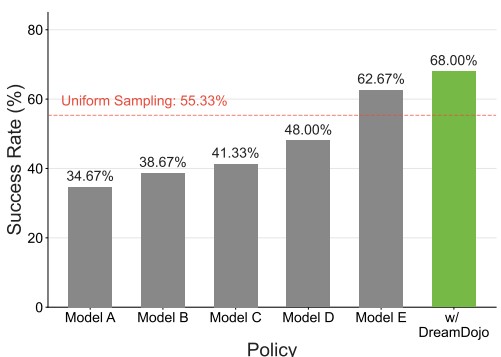

*Figure 4.* **Real vs. DreamDojo success rates.**

*Table 3.* **Effects of different action conditioning methods.** Latent action conditioning performs on par with the ideal settings in simulation quality and is the most scalable in use. We denote retargeted action and MANO in gray because they represent ideal collection setups when additional action capture devices are equipped.

| Method | In-lab Eval | | |
|---|---|---|---|
| | PSNR↑ | SSIM↑ | LPIPS↓ |
| w/o pretrain | 20.576 | 0.774 | 0.222 |
| action-free | 20.797 | 0.773 | 0.222 |
| latent action | 20.913 | 0.776 | 0.219 |
| retargeted action | 20.960 | 0.773 | 0.219 |

| Method | EgoDex Eval | | |
|---|---|---|---|
| | PSNR↑ | SSIM↑ | LPIPS↓ |
| w/o pretrain | 19.952 | 0.787 | 0.219 |
| action-free | 19.924 | 0.783 | 0.222 |
| latent action | 20.344 | 0.790 | 0.214 |
| MANO | 20.474 | 0.795 | 0.211 |

*Table 4.* **Results of our distillation pipeline.** Our distilled model is capable of real-time inference (10.93 FPS) with minor degradation in long-horizon rollouts and performance close to that of the teacher.

| Method | GR-1 Long Eval | | | |
|---|---|---|---|---|
| | PSNR↑ | SSIM↑ | LPIPS↓ | FPS↑ |
| Teacher | 14.086 | 0.442 | 0.412 | 2.72 |
| Student | 13.146 | 0.379 | 0.485 | 10.93 |

*Figure 5.* **Model-based planning results.**

et al., 2024), we utilize Pearson correlation coefficient to quantify linear agreement between real-world and Dream-Dojo's success rates and Mean Maximum Rank Violation (MMRV) to measure the rank consistency between these two. Figure 4 shows that DreamDojo's success rate has a strong linear correlation with real-world success rate (Pearson $r$=0.995) and maintains a highly consistent ranking (MMRV=0.003), indicating that DreamDojo is able to serve as a reliable simulator for policy evaluation.

**Model-based planning.** Being able to simulate future outcomes conditioned on actions allows model-based planning that can strengthen and correct the policies at test time (Qi et al., 2025). In this paper, we adopt a simple algorithm for model-based planning. Specifically, similar to the policy evaluation experiment above, we setup 10 AgiBot fruit packing scenes as our touchstone. We ensemble 5 model checkpoints from training to generate action proposals that

exhibit sufficient variance at inference time. These action proposals are sent to DreamDojo to predict future video trajectories. Subsequently, the best proposal is selected by an external value model that takes a short video clip as input and is executed by the robot. The implementation details of the value model are provided in the Appendix.

To ensure execution efficiency, we batch all the action inputs and process them using the distilled DreamDojo-2B. While our world model will introduce additional latency, it will also significantly improve the overall performance. The results are presented in Figure 5. With the help of DreamDojo, the policies can anticipate the outcomes of their predictions in advance and adaptively select the most promising mode for execution, improving the success rate by 5.33% over the best model checkpoint. Compared to uniformly sampling from all policy proposals, applying model-based planning with DreamDojo yields nearly a 13% increase in success rate. These results highlight the huge promise of DreamDojo for online policy steering.

**Live teleoperation.** We can also provide action conditions by connecting the teleoperative devices used for robot data collection to DreamDojo. To verify this, we deploy DreamDojo-2B on a local desktop equipped with a 5090 GPU and connect a PICO VR controller to capture the upper-body action inputs for the G1 robot. As a result, we found that we could directly teleoperate the virtual robot at real-time speed. An example is shown in Figure 6. For live teleoperation videos, please visit our website.

## 5. Conclusion

In this paper, we introduce DreamDojo, a generalist world model that can simulate dexterous robotics tasks and generalize to unseen scenarios. Our model is pretrained on large-scale human datasets that encompass a wide variety of daily interactions. To further enhance knowledge transfer and action controllability, continuous latent actions are introduced as proxy actions for all videos. We further introduce a distillation pipeline that enables stable long-horizon interactions at real-time speed. Extensive evaluations underscore the significance of DreamDojo, demonstrating improved physics understanding and action following in out-of-distribution scenarios, a positive correlation with real-world evaluations, and real-time interactivity for live teleoperation and test-time policy steering. We hope our effort can pave the way for general-purpose robot world models.

## Impact Statement

This paper presents work whose goal is to advance the field of machine intelligence. There are many potential societal consequences of our work, none which we feel must be specifically highlighted here.

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

## A. Limitations

While DreamDojo demonstrates significant improvements over the baseline, it is by no means perfect when simulating uncommon actions, such as slapping and fast waving. Additionally, when conducting policy evaluation, the absolute success rates in DreamDojo are often higher than their real counterparts, indicating a limitation in accurately generating nuanced failures. Future work should explore how to cover broader action distribution, *e.g.*, using policy rollouts (Zhu et al., 2025c; Ho et al., 2025). We also believe that there remains significant space for improving inference speed through engineering optimizations (Hong et al., 2025; Parker-Holder & Fruchter, 2025; Team et al., 2026). In addition, it does not naturally support multi-view simulation, which is crucial for state-of-the-art policies (Bjorck et al., 2025; Black et al., 2024). Moreover, how to retain the pretrained knowledge as much as possible has not been studied in depth. Future work could explore other fine-tuning strategies (Hu et al., 2022; Yadav et al., 2025) to achieve better post-training performance on the target embodiment.

## B. Experiment Details

**Training and inference.** The latent action model is a 700M spatiotemporal Transformer (Bruce et al., 2024) that is trained for 400k steps with a total batch size of 256. The dimension of the latent action is 32. The model has 24 encoder blocks for latent action extraction and 24 decoder blocks for forward dynamics prediction. It is trained on a data mixture of the three human video datasets, as well as our in-house robot datasets, including Unitree G1, Fourier GR-1, AgiBot, and YAM. The original videos are temporally downsampled by a random factor of $\{1, 2, 3, 4\}$ to capture various kinds of motions. The video frames are center cropped and resized to a fixed resolution of $240 \times 320$. The $\beta$ in Equation (3) is set to $10^{-6}$ to achieve a good trade-off between representation capacity and transferability for post-training. We employ the AdamW (Loshchilov & Hutter, 2019) with a weight decay of 0.01 and a constant learning rate of $2.5 \times 10^{-5}$ to train the latent action model from scratch.

The world model is initialized from Cosmos-Predict2.5 (Ali et al., 2025) and pretrained on the mixture of our In-lab, EgoDex, and DreamDojo-HV datasets, with a sampling ratio of 1:2:10, respectively. The video frames are center cropped and resized to a fixed resolution of $640 \times 480$, and then clipped into sequences with a length of 13 for pretraining. The text condition is fixed as an empty prompt. We present two variants of DreamDojo: a 2B model and a 14B model. Both models are pretrained for 140k steps with an effective batch size of 1024 using 256 H100 GPUs. We use AdamW (Loshchilov & Hutter, 2019) with a weight decay of 0.1, and set the learning rate to $1.6 \times 10^{-4}$. An exponential moving average (EMA) is maintained throughout the training and used to generate all our results.

In the post-training stage, the videos of the target embodiment (*e.g.*, G1, GR-1, AgiBot) are sampled at roughly 10 Hz to capture sufficient motions. The video clips are then organized as sequences with a length of 13. The first frame serves as the condition frame, and the raw actions are processed as relative actions with a length of 12. The world model is finetuned with all weights updated using a similar hyperparameter setting as in the pretraining stage. By default, post-training is conducted with 128 H100 GPUs for 50k steps with a batch size of 512.

The distillation stage initializes the autoregressive student model with the weights of the teacher, while replacing bidirectional attention with causal attention over a sliding window size of 12 frames. First, for the warmup stage, we generate 10k ODE trajectories and train for 10k iterations. Next, for the distillation step, we have the student randomly generate between 13 and 49 frames during training, and compute loss on the last 13 generated frames. We run this distillation step for 3k iterations. All distillation is conducted on 64 H100 GPUs, using a batch size of 256 for the warmup stage and 64 for the distillation stage.

During inference, the teacher model utilizes 35 denoising steps for generation, while the distilled model reduces this number to 4 steps. Classifier-free guidance (Ho & Salimans, 2022) is disabled as we empirically found it brings limited benefits.

**Evaluation protocol.** To quantify the model performance on our evaluation sets, we use PSNR (Hore & Ziou, 2010), SSIM (Hore & Ziou, 2010), and LPIPS (Zhang et al., 2018) as three main automatic metrics. When evaluating the models without distillation, we generate 100 future videos over three rounds by autoregressively resetting the condition frame with the last prediction to make the discrepancies between different variants more discriminative, resulting in 100 samples with 49 frames for most of our evaluations. We choose Fourier GR-1 as a representative target embodiment for most ablative studies.

Since the ground-truth videos are unavailable for EgoDex-novel Eval and DreamDojo-HV-novel Eval, we design a human

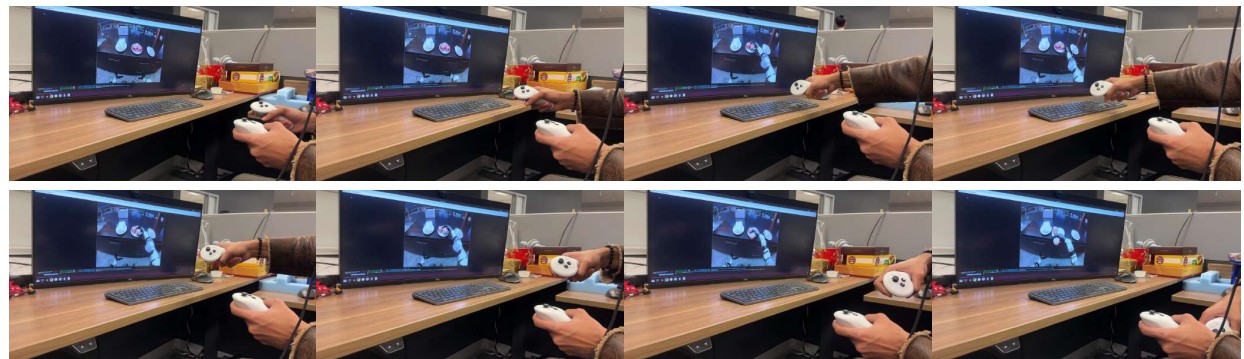

*Figure 6.* **Live teleoperation.** We can teleoperate a virtual G1 robot using the PICO VR controller in real time.

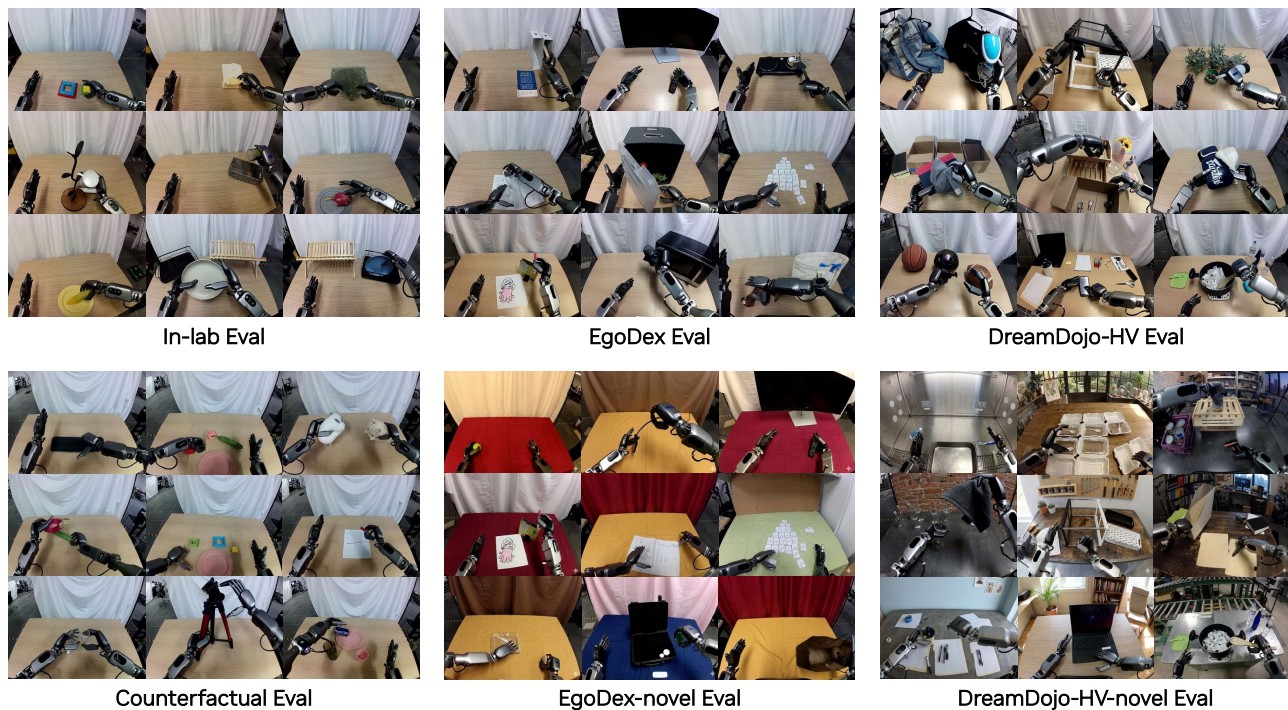

In-lab Eval  EgoDex Eval  DreamDojo-HV Eval

Counterfactual Eval  EgoDex-novel Eval  DreamDojo-HV-novel Eval

*Figure 7.* **Benchmark visualization.** We rigorously construct six evaluation benchmarks that reflect the diverse scenarios and actions present in human datasets, while being out-of-distribution for the robot training datasets.

preference evaluation protocol following recent advances (Gao et al., 2024; Wan et al., 2025; Yin et al., 2025). Specifically, we make a web UI and invite 12 volunteers to judge side-by-side video pairs from physics correctness of object interactions and action following compared to the ground-truth video. For physics correctness, they are suggested to focus on object permanence, shape consistency, and contact causality. For action following, we encourage the evaluators to pay more attention to the pose of the robots and allow for a "tie". We also provide evaluation examples beforehand to justify the key factors the evaluators should focus on. The order of the videos in each pair will be randomly switched to avoid bias, and we average all win rates against the anchor model to obtain the final results. Figure 8 shows our web UI for our human preference evaluation, allowing the evaluators to assess physics correctness and action following intuitively.

**Policy evaluation details.** We set up 20 different scenes for the fruit packing tasks, covering various combinations of multiple fruits (pear, mango, banana, and starfruit) located in different places on the table. The success rate is determined by the number of fruits successfully picked up from the table and placed into the bag, with 5 fruits designated as 100% success. For each scene, we collect an approximately 80-second rollout in the real world and simulate the entire rollout with the post-trained DreamDojo-2B using the same initial frame. The generated rollouts are scored by human evaluators based on consistent criteria as the real world. The final success rate is averaged across all 20 scenes for both real-world and DreamDojo.

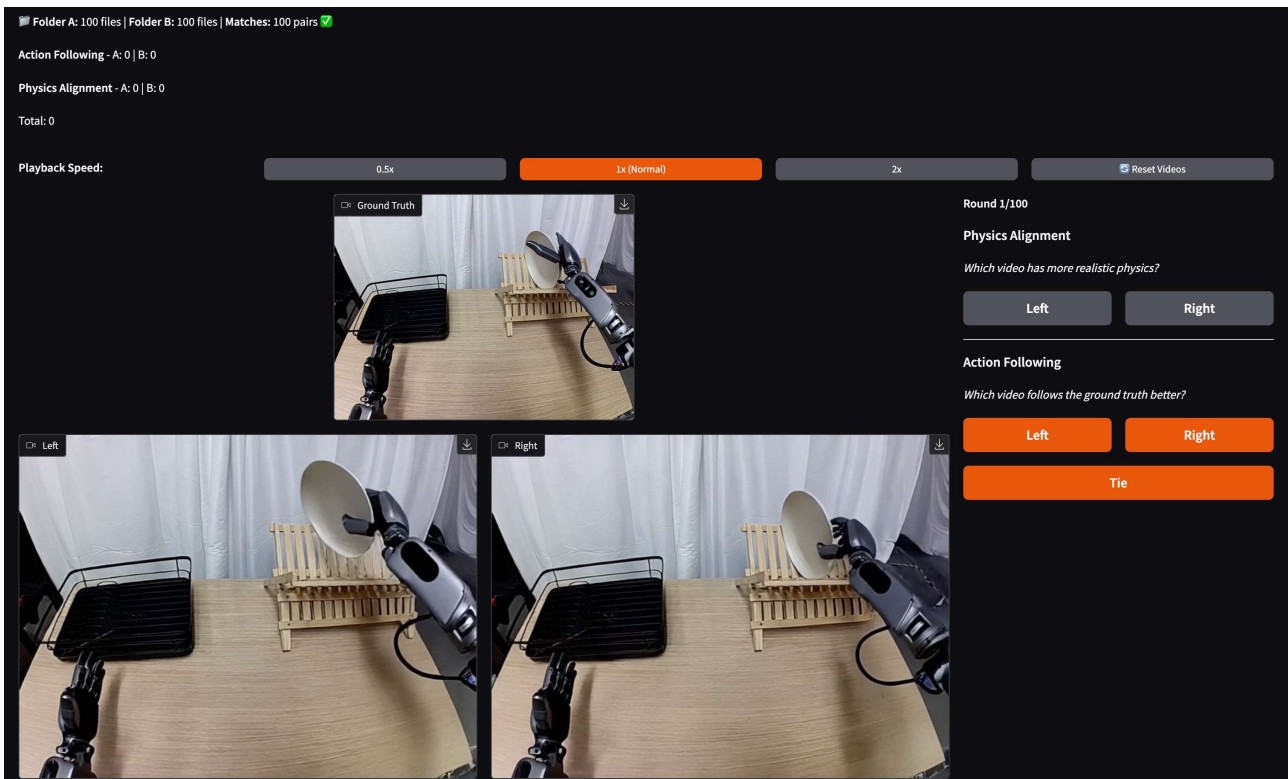

*Figure 8.* **Web UI for human preference evaluation.** To intuitively compare physics correctness and action controls, we devise a web UI that can display the ground-truth video alongside two videos generated by two different models simultaneously. The order of the generated videos will be randomly switched to avoid any potential bias.

## C. Related Work

**World model.** World models can simulate world transitions in response to actions, which have been proven critical for developing intelligent agents (Ha & Schmidhuber, 2018; Hafner et al., 2025a; Alonso et al., 2024; Richens et al., 2025). Building upon advances in generative models, a surge of works have developed high-quality video world models for simulating interactive games (Yu et al., 2025; Kim et al., 2020; Alonso et al., 2024; Guo et al., 2025a; Parker-Holder et al., 2024; Parker-Holder & Fruchter, 2025; Ye et al., 2025a; Sun et al., 2025a; Kanervisto et al., 2025; Hafner et al., 2025b) and autonomous driving (Kong et al., 2025; Kim et al., 2021; Hu et al., 2023a; Russell et al., 2025; Yang et al., 2024a; Bar et al., 2025). Motivated by successes in these domains, recent works have also introduced video generative models to simulate robot manipulation tasks (Wu et al., 2024; Wang et al., 2025a; Li et al., 2025c; Zhu et al., 2025a; Jiang et al., 2025a; Guo et al., 2025b), which hold great promise for scalable policy evaluation (Li et al., 2025d; Quevedo et al., 2025; Zbinden et al., 2025; Tseng et al., 2025; Team et al., 2025), reinforcement learning (Yang et al., 2024b; Xiao et al., 2025; Li et al., 2025a; Jiang et al., 2025b; Zhu et al., 2025c; Zhang et al., 2025a), and policy steering (Zhou et al., 2025; Assran et al., 2025; Jain et al., 2025; Du & Song, 2025; Wu et al., 2025; Qi et al., 2025). However, existing models are typically trained and evaluated in in-distribution settings, leaving it unclear whether these models can truly facilitate planning in unseen scenarios.

Another thread of research focuses on world model pretraining from internet-scale videos to improve downstream performances (Seo et al., 2022; Mendonca et al., 2023; Wu et al., 2023; Zhang et al., 2024). More related to ours, VAP (Wang et al., 2025b) utilizes 2D skeletons to unify action conditions from different robots and hands for joint training. AdaWorld (Gao et al., 2025) and the follow-up works (Wang et al., 2025c; Garrido et al., 2026) propose pretraining a world model with latent actions to enhance transferability. Additionally, DexWM (Goswami et al., 2025) leverages human videos to help generalization to unseen dexterous manipulation skills. However, they primarily focus on tabletop datasets in laboratory setups and demonstrate inferior visual quality compared to recent advancements. In contrast, we introduce the first generalist world model for dexterous manipulation, which exhibits strong generalization in simulating diverse out-of-distribution manipulation skills across multiple embodiments.

**Latent action.** Internet-scale video is an intriguing source for sparking emergent abilities (Yang et al., 2024c; Wiedemer

et al., 2025), but the unavailability of action labels could significantly hinder learning efficiency. To address this issue, latent actions have recently been proposed as a promising approach for learning from unlabeled videos (Schmidt & Jiang, 2024; Ye et al., 2025b; Jang et al., 2025; Bu et al., 2025b;a). Besides serving as supervision for policy learning, latent actions can also be utilized as a control interface for world models (Bruce et al., 2024; Chen et al., 2024; Gao et al., 2025; Wang et al., 2025c; Garrido et al., 2026). Several works have also demonstrated the effectiveness of continuous latent actions (Gao et al., 2025; Liang et al., 2025; Yang et al., 2025b; Liu et al., 2025b). Inspired by these explorations, we extract latent actions as a unified proxy for our generalist world model and investigate how this approach can promote robust generalization when interacting with unseen objects after adapting to new embodiments.

**Autoregressive video generation.** Autoregressive video generation offers the finest granularity and flexibility for interaction (Weng et al., 2024), which is well-suited for action-conditioned world modeling (Valevski et al., 2025; Bruce et al., 2024; Gao et al., 2025; Huang, 2025). To accelerate inference speed, previous methods (Yin et al., 2025; Lin et al., 2025) distill the bidirectional model into an autoregressive student model that only requires fewer steps to generate videos of comparable quality. More recently, Self Forcing and its follow-ups (Huang et al., 2025b; Liu et al., 2025a; Cui et al., 2025; Shin et al., 2025; Zhang et al., 2025b) further reduce long-term drift by mirroring the inference process during training. Building upon these techniques, we present a distillation pipeline that significantly boosts inference speed to real-time levels while ensuring the final model is aware of historical context. This makes our distilled model readily applicable to various downstream tasks, such as performing long-horizon dexterous teleoperation in real time.

## D. Additional Results

### D.1. Effects of Our Data Mixtures

In addition to the quantitative evaluations, we demonstrate the effectiveness of human data pretraining through visualizations. The samples in Figure 9 verify that incorporating human data pretraining is essential for precise physics modeling of objects not captured by the robot dataset.

### D.2. Comparison with Prior Robot World Models

To contextualize DreamDojo against the state-of-the-art robot world model, we additionally post-train Ctrl-World (Guo et al., 2025b) using its official configuration and compare it against DreamDojo-2B on all four evaluation sets. As shown in Table 5, DreamDojo-2B surpasses Ctrl-World by a large margin across all evaluation sets in both PSNR and SSIM, demonstrating that the proposed human video pretraining recipe yields substantially better generalization than prior dedicated robot world models.

*Table 5.* **Comparison with Ctrl-World.** DreamDojo-2B consistently outperforms the prior state-of-the-art Ctrl-World across all four evaluation sets.

| Method | In-lab Eval | | | EgoDex Eval | | | DreamDojo-HV Eval | | | Counterfactual Eval | | |
|---|---|---|---|---|---|---|---|---|---|---|---|---|
| | PSNR↑ | SSIM↑ | LPIPS↓ | PSNR↑ | SSIM↑ | LPIPS↓ | PSNR↑ | SSIM↑ | LPIPS↓ | PSNR↑ | SSIM↑ | LPIPS↓ |
| Ctrl-World | 20.075 | 0.719 | 0.215 | 19.637 | 0.735 | 0.205 | 17.920 | 0.694 | 0.231 | 19.391 | 0.736 | 0.216 |
| DreamDojo-2B | 21.114 | 0.774 | 0.222 | 20.411 | 0.775 | 0.226 | 18.813 | 0.747 | 0.238 | 20.907 | 0.787 | 0.192 |

### D.3. Effects of Including Failure Rollouts in Post-Training

A key limitation of training only on expert demonstrations is that the model may not learn to simulate nuanced failure trajectories. To examine whether incorporating policy rollouts at the post-training stage helps, we post-train two variants of DreamDojo on the bimanual YAM dataset: one variant uses 35 hours of demonstration data, while the other combines the 35 hours of demonstration data with 5 hours of policy rollouts (of which approximately $5/6$ are failures). Both variants are trained for 20k steps, and the mixture ratio of demonstration to rollout is 2:1 in the second variant. We then evaluate the PSNR of both variants on a held-out demonstration set (Success) and a held-out failure rollout set (Failure). As shown in Table 6, including failure rollouts in post-training yields a small but consistent gain on the success set, and a much larger gain on the failure set, indicating that the model becomes substantially better at simulating unsuccessful trajectories without sacrificing simulation quality on successful ones.

*Table 6.* **Effects of including failure rollouts in post-training.** Mixing a small amount of policy rollouts into post-training substantially improves the simulation of failure trajectories, with no degradation on successful ones.

| Eval Set | w/o rollout | w/ rollout |
|---|---|---|
| Success | 21.522 | 21.547 |
| Failure | 20.656 | 22.220 |

## D.4. Sensitivity to Post-Training Data Scale

To understand how the amount of post-training data interacts with our human video pretraining, we vary the GR-1 post-training data scale and finetune two variants for 15k steps each: one initialized from our pretrained DreamDojo, and the other from Cosmos-Predict2.5 without our human video pretraining. Both variants are evaluated on the In-Lab Eval set using PSNR. As shown in Table 7, models with our pretraining outperform those without across all scales. Notably, our pretraining makes the model significantly less sensitive to the size of the post-training dataset; in contrast, the variant without our pretraining performs noticeably worse at 8h, suggesting an overfitting tendency on small post-training corpora. This indicates that our human video pretraining substantially reduces the post-training data requirement for adapting to a new embodiment.

*Table 7.* **Sensitivity to post-training data scale.** Our pretraining consistently outperforms training without pretraining and makes the model robust to small post-training corpora.

| Data Scale | w/o pretrain | w/ pretrain |
|---|---|---|
| 8h | 18.889 | 21.261 |
| 44h | 19.081 | 21.175 |
| 88h | 19.074 | 21.109 |

## D.5. Pretraining Benefits Propagating to Downstream Planning

To verify whether the benefits of our human video pretraining can propagate to downstream applications, we additionally distill a model post-trained from Cosmos-Predict2.5 on AgiBot fruit packing (i.e., without our human video pretraining) and compare it against our distilled DreamDojo-2B under the model-based planning setup in Section 4.5. We also include a baseline that uniformly samples from the policy action proposals without any world-model-based selection. As shown in Table 8, both world models help over the unified sampling baseline (37%), but the model with our pretraining is substantially better at selecting the correct actions, achieving a 70% success rate compared to 43% for the variant without pretraining. This indicates that the generalization benefits acquired from large-scale human videos translate directly into better downstream planning performance.

*Table 8.* **Pretraining benefits in model-based planning.** Our human video pretraining propagates to substantially higher downstream success rates in model-based planning on AgiBot fruit packing.

| unified sampling | w/o pretrain | w/ pretrain |
|---|---|---|
| 37% | 43% | 70% |

## D.6. Generalization to Unseen Scenarios

To benchmark the generalization ability in unseen scenarios, we generate video samples using the two final models, DreamDojo-2B and DreamDojo-14B, and conduct evaluations with Cosmos-Predict2.5 without human video pretraining. All models are post-trained for 30k steps. After post-training, we ask 12 volunteers to evaluate three model pairs: DreamDojo-2B *vs.* Cosmos-Predict2.5, DreamDojo-14B *vs.* Cosmos-Predict2.5, and DreamDojo-14B *vs.* DreamDojo-2B. The evaluation is conducted on 50 samples from EgoDex-novel Eval and DreamDojo-HV-novel Eval. The results in Table 9 show that our DreamDojo-2B surpass the original Cosmos-Predict2.5 in both physics correctness and action following by a non-trivial margin, while DreamDojo-14B exhibits a clear advantage over DreamDojo-2B in both axes due to its large capacity.

## D.7. Effects of Our Model Designs

We provide qualitative comparisons of the results from different model designs as outlined in Table 10. Both relative actions and chunked injection significantly enhance simulation quality, underscoring their importance for achieving precise action

*Table 9.* **Human preference evaluation in diverse out-of-distribution scenarios.** DreamDojo outperforms the pretrained Cosmos-Predict2.5 by a non-trivial margin. Our DreamDojo-14B demonstrates the most competitive performance in both physics correctness and action following.

| Metric | DreamDojo-2B > Cosmos-Predict2.5 | DreamDojo-14B > Cosmos-Predict2.5 | DreamDojo-14B > DreamDojo-2B |
|---|---|---|---|
| Physics Correctness | 62.50% | 73.50% | 72.50% |
| Action Following | 63.45% | 72.55% | 65.53% |

*Table 10.* **Ablations of architecture and loss designs.** Our design choices can effectively enhance the simulation quality of both expert and counterfactual trajectories.

| Modifications | | | GR-1 Val | | | Counterfactual Eval | | |
|---|---|---|---|---|---|---|---|---|
| relative | chunked | temporal | PSNR↑ | SSIM↑ | LPIPS↓ | PSNR↑ | SSIM↑ | LPIPS↓ |
| | | | 16.199 | 0.557 | 0.315 | 19.448 | 0.768 | 0.211 |
| ✓ | | | 16.522 | 0.576 | 0.304 | 19.482 | 0.772 | 0.212 |
| ✓ | ✓ | | 17.626 | 0.620 | 0.267 | 20.783 | 0.790 | 0.193 |
| ✓ | ✓ | ✓ | 17.630 | 0.622 | 0.266 | 20.980 | 0.796 | 0.189 |

*Table 11.* **Generalization ability after distillation.** DreamDojo shows consistently better generalization than the baseline.

| Method | In-lab Eval | | | EgoDex Eval | | | DreamDojo-HV Eval | | | Counterfactual Eval | | |
|---|---|---|---|---|---|---|---|---|---|---|---|---|
| | PSNR↑ | SSIM↑ | LPIPS↓ | PSNR↑ | SSIM↑ | LPIPS↓ | PSNR↑ | SSIM↑ | LPIPS↓ | PSNR↑ | SSIM↑ | LPIPS↓ |
| w/o pretrain | 20.304 | 0.770 | 0.230 | 19.119 | 0.762 | 0.240 | 17.869 | 0.736 | 0.259 | 19.782 | 0.758 | 0.232 |
| w/ pretrain | 20.733 | 0.782 | 0.220 | 19.313 | 0.765 | 0.235 | 18.195 | 0.740 | 0.254 | 19.891 | 0.746 | 0.234 |

controllability. Additionally, the proposed temporal consistency loss further reinforces the modeling quality of objects.

### D.8. Ablations of Our Design Choices

To efficiently verify the effectiveness of our architectural design and training objective, we finetune Cosmos-Predict2.5 for 30k steps only on the GR-1 training dataset. The finetuned models are then evaluated on a held-out GR-1 validation set with expert demonstrations and the Counterfactual Eval set. Starting with the simple Cosmos-Predict2.5 base architecture, we gradually apply our modifications: relative action transformation, chunked action injection, and the temporal consistency loss. The evaluation results are shown in Table 10. Both relative actions and chunked injection can significantly improve simulation quality, indicating their importance for achieving precise action controllability. The proposed temporal consistency loss further improves performance on both benchmarks, demonstrating its effectiveness in enhancing action following and object modeling. In Figure 10, we also provide qualitative comparisons for these variants.

### D.9. Benefits of Distillation

We ablate the choice of teacher model in Table 11, evaluating the distillation results of a teacher pretrained on human videos versus one without pretraining (Cosmos-Predict2.5). Across all four evaluation datasets, we observe that the former significantly outperforms the latter. This suggests that the benefits of generalization achieved through our human video pre-training are preserved after distillation, resulting in student models that also excel in unseen scenarios.

We compare the 1-minute videos generated by the teacher model and the student model in Figure 11. DreamDojo can continuously predict long-horizon rollouts in real time with strong stability and action following.

In Figure 12, we also visualize representative samples that illustrate the superior consistency of the student model. The distilled DreamDojo can recover objects from occlusions by modeling a short context, whereas the teacher model is unable to achieve this due to its single-frame conditioning.

### D.10. DreamDojo-HV Samples

To assist a better understanding of the DreamDojo-HV dataset, we visualize more data samples in Figure 13, highlighting its extensive coverage of interaction types and scenarios.

### D.11. PSNR Curves in Post-Training

We visualize how the PSNR scores will evolve during post-training. From Figure 14, we notice that pretraining with latent actions can reach a much higher upper bound than action-free pretraining and without pretraining, especially on EgoDex

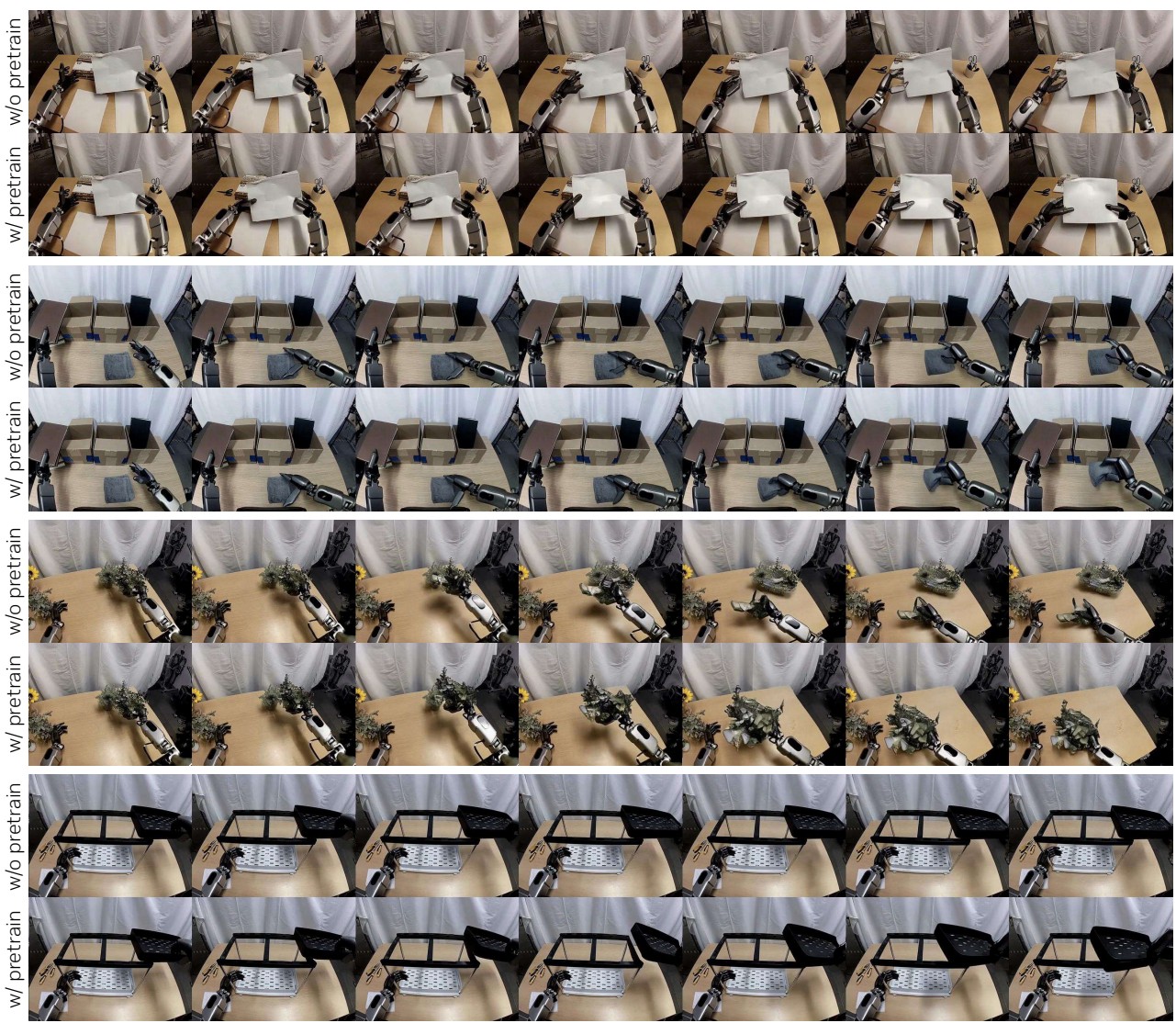

*Figure 9.* **Qualitative comparison of human data pretraining effects.** Through pretraining on diverse human interaction data, DreamDojo acquires a generalizable understanding of general physics, resulting in more realistic simulation for objects that are unseen in the target robot dataset.

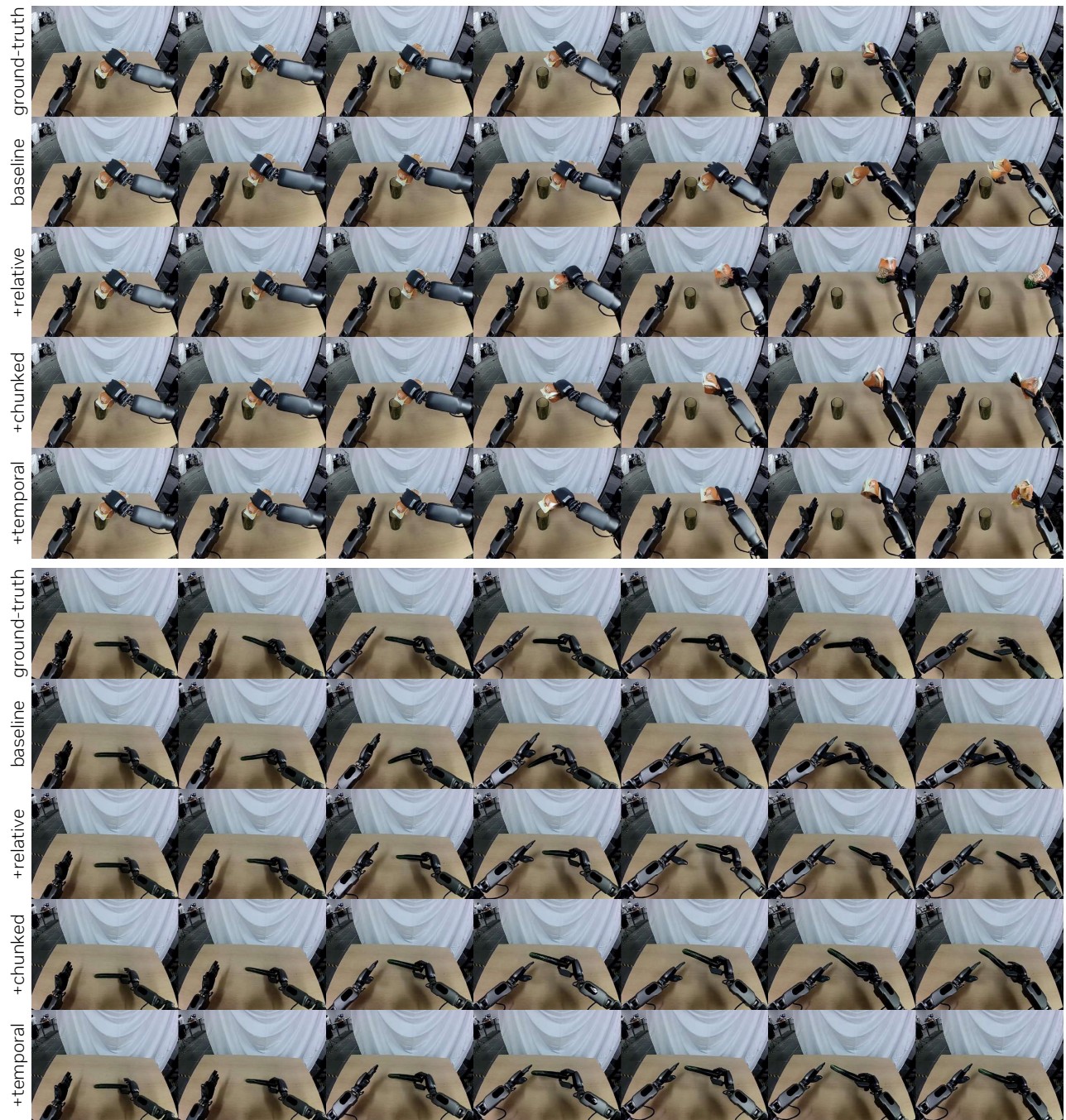

*Figure 10.* **Qualitative comparison of our design choices.** Applying all our techniques results in the best capabilities for object modeling and action following.

Eval.

## D.12. Value Model

To automatically judge the value of the predicted futures, we train an external model based on the DINOv2 (Oquab et al., 2023) architecture. Our value model takes a video clip consisting of 4 frames as input. The DINOv2 backbone is frozen and independently extracts image features from each frame. The features from all frames are then processed by a value prediction module with global attention. The value prediction module is trained to estimate the number of time steps remaining until each subtask boundary, which is defined as the keyframe between consecutive subtask language annotations. The ground-truth value is normalized by the maximum subtask interval in the dataset. For each generated video, the value estimation is performed as a sliding window with a stride of 1. The final value of the current video is defined as the average of all clips from the start to the dip before the estimated value increases. The action proposal with the lowest value (*i.e.*, closest to subtask completion) will be selected for real-world execution. A visualization of the accuracy of our value model is shown in Figure 15.

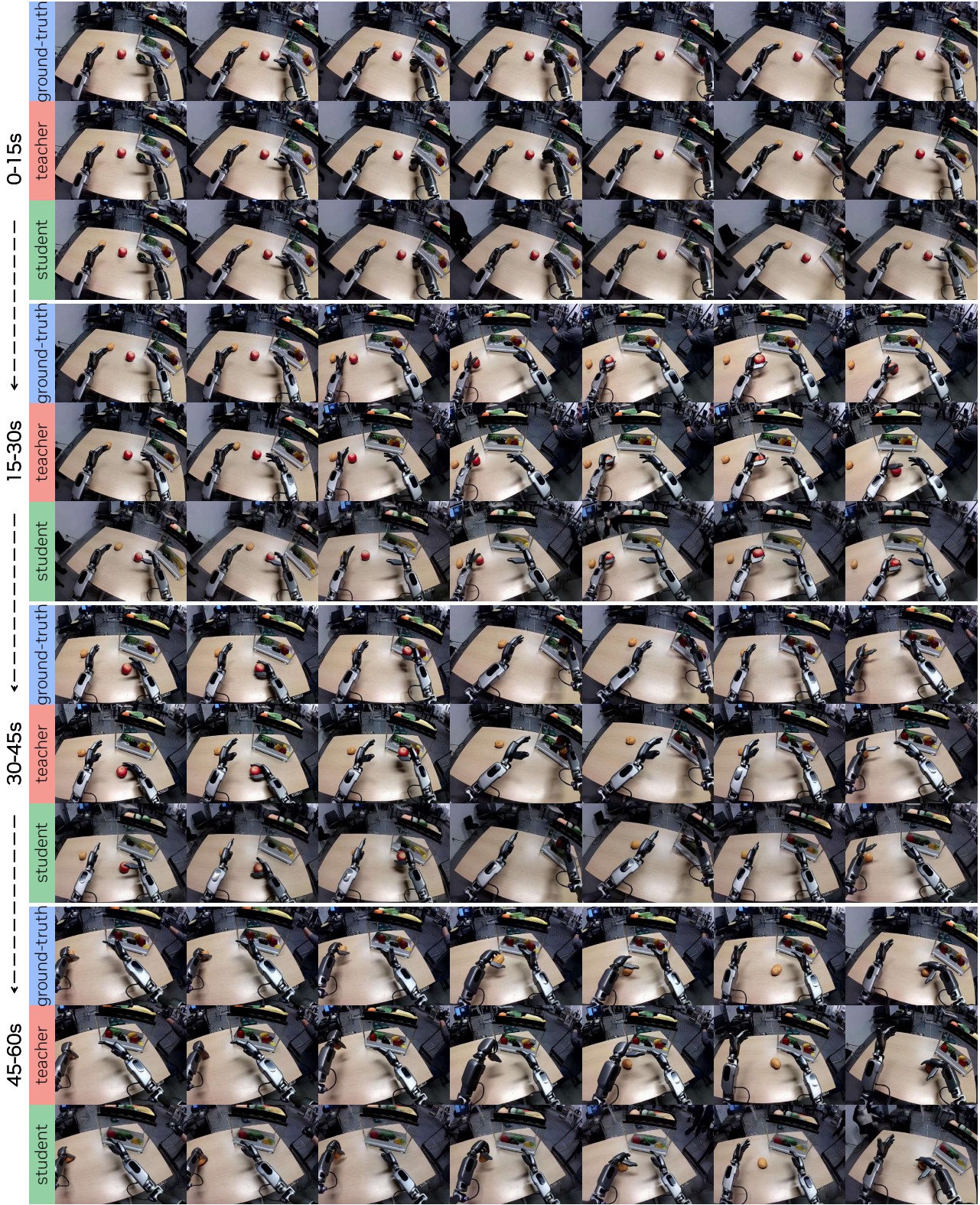

*Figure 11.* **Long-horizon rollouts for 1 minute.** Note that the teacher model generates videos in a chunk-wise manner and operates at a speed (2.72 FPS) that is 4× slower than that of the student model (10.93 FPS).

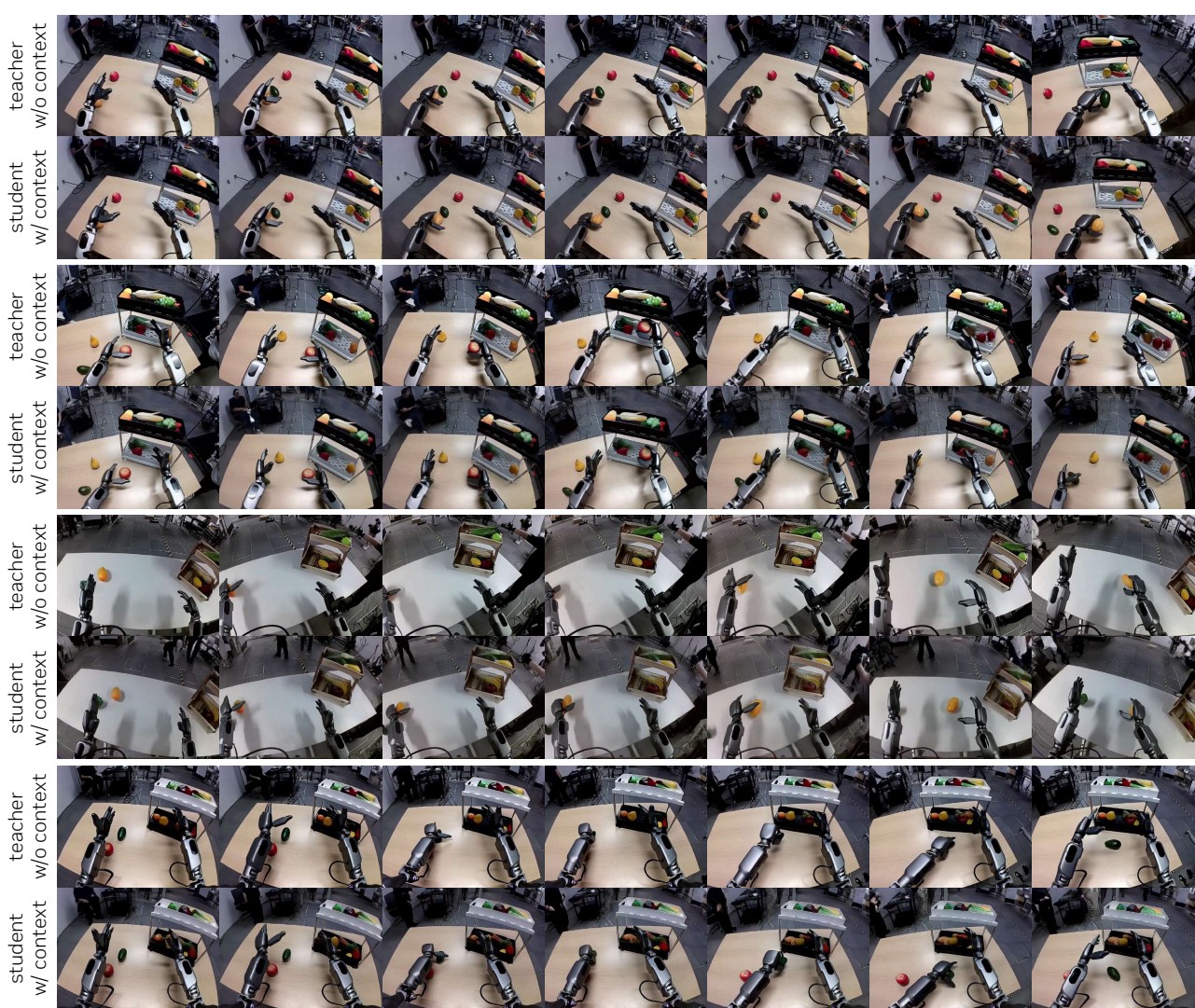

*Figure 12.* **The advantage of student context.** The student model exhibits better consistency in handling occlusions and camera shifts, while the teacher model has no way to ensure that due to the missing context.

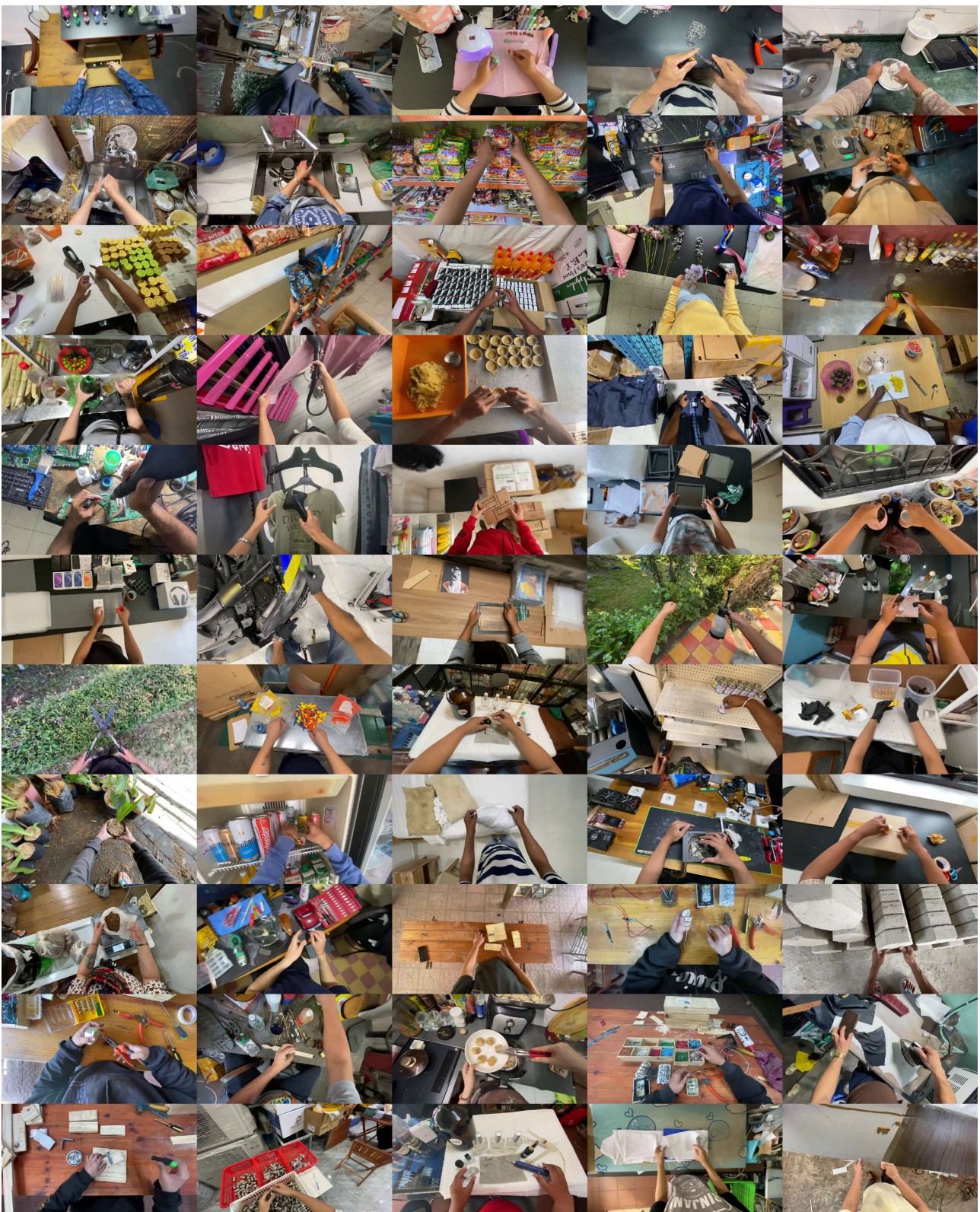

*Figure 13.* **Diversity of DreamDojo-HV.** We visualize more samples from the curated DreamDojo-HV dataset, which encompasses extremely diverse actions and tool-using scenarios.

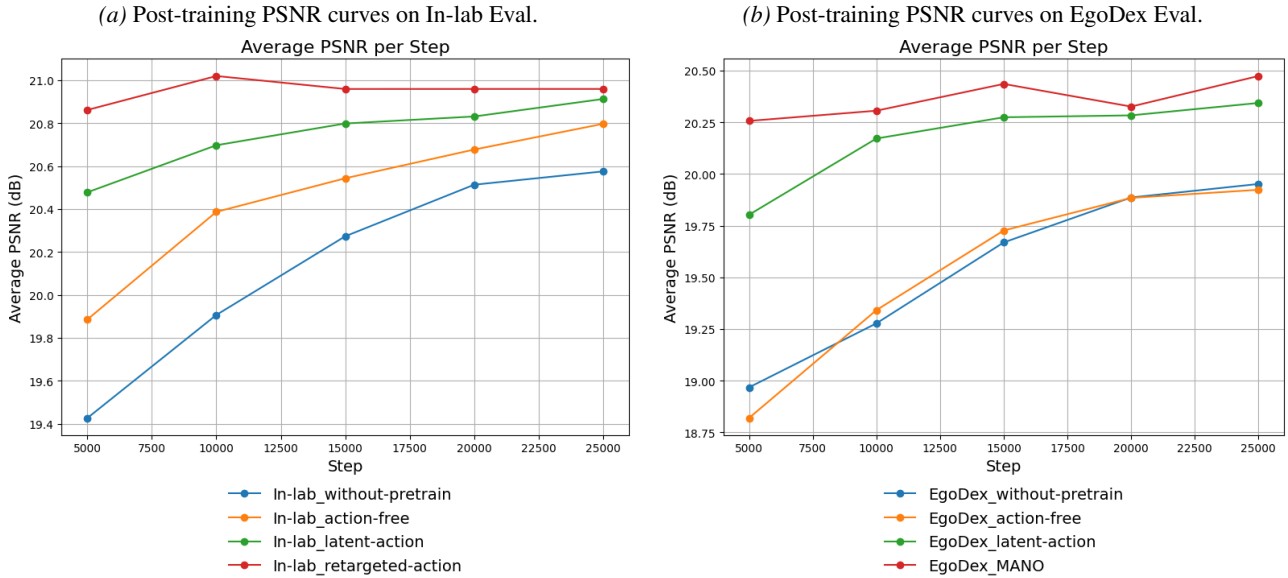

*(a)* Post-training PSNR curves on In-lab Eval.

*(b)* Post-training PSNR curves on EgoDex Eval.

*Figure 14.* **Post-training PSNR curves using different action conditioning.** Latent action conditioning can achieve comparable performance as high-quality action labels obtained using extra devices. On EgoDex Eval, our approach can also reach a much higher upper bound compared to action-free pretraining and without pretraining.

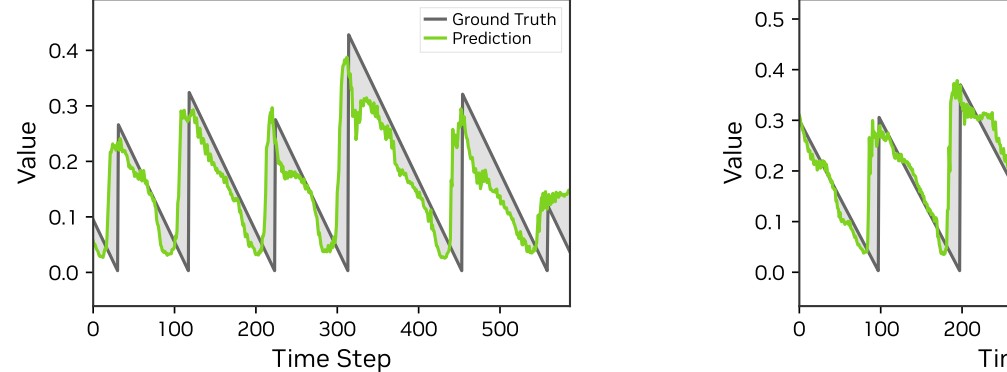

*Figure 15.* **Value model estimation.** We visualize the estimated value and the ground-truth value of two representative episodes. Our value model reliably estimates the number of steps remaining to complete the current subtask.

