# OpenReview forum: "DreamDojo: A Generalist Robot World Model from Large-Scale Human Videos"
_ICML.cc/2026/Conference — ICML 2026 spotlight_

### Official Review · Reviewer_KAD9 · 2026-03-10

**Soundness:** 3
**Presentation:** 3
**Significance:** 3
**Originality:** 2
**Overall Recommendation:** 5
**Confidence:** 4

**Summary:**

This paper presents DreamDojo, a world model for dexterous robotic tasks that is pretrained on 44k hours of egocentric human videos. To address the scarcity and heterogeneity of robot action labels, the paper introduces continuous latent actions as a unified proxy action space, enabling action-conditioned world modeling from large-scale unlabeled videos. The paper also proposes a distillation pipeline to improve efficiency and long-horizon consistency, enabling real-time autoregressive prediction. Experiments on multiple out-of-distribution benchmarks and downstream settings, such as teleoperation, policy evaluation, and planning, are used to demonstrate the potential of large-scale video-pretrained world models for open-world robot interaction.

**Compliance With Llm Reviewing Policy:**

Affirmed.

**Final Justification:**

This paper presents a well-engineered world model with strong generalization and real-time inference. Originality is somewhat limited as the pipeline combines known techniques, but the scale and systematic validation are valuable. The rebuttal resolved my concerns with additional baselines and action-following evaluations. My score has risen from 4 to 5.

**Key Questions For Authors:**

1. Could you provide comparisons with more robot world model baselines?
2. Some experiments on the use of DreamDojo as world-model-based RL for VLA post-training would be more convincing.
3. Some metrics for measuring how well the predicted frames are aligned with the input actions are missing; the current paper only discusses video quality.

**Limitations:**

yes

**Strengths And Weaknesses:**

**Strengths**:
- This paper presents a strong robot world model, with strong performance and fast inference speed, and the evaluation is conducted on both video generation quality and downstream robotic tasks such as policy evaluation and model-based planning.
- This paper also presents a large-scale video dataset, which is beneficial to the community.

**Weaknesses**:
- The main components of the paper, such as video pretraining, latent actions, and the distillation process, have been well examined in prior work, and the overall pipeline is a composition of existing methods.
- The baselines in the experimental section are relatively limited and only include comparisons with Cosmos-Predict, while missing other recent robot world models.

---

> ### Author Rebuttal · Authors · 2026-03-31
>
> Thanks for the feedback. We address each concern below and will include all results in the revision.
>
> **(W1) The main components have been examined in prior work.** While our work is inspired by several previous works, we are the first to curate such a large-scale dataset and systematically validate all designs at this scale. To the best of our knowledge, we present the first model of its kind that demonstrates exceptional generalization ability and can perform long-horizon rollout at a real-time speed. We will open-source the whole training pipeline and hope it will pave the way for general-purpose robot world models.
>
> **(W2&Q1) Missing other recent robot world model baselines.** We focus on comparing with Cosmos to demonstrate the effects of the proposed pretraining recipe with minimal architectural differences. To address the reviewer’s concern, we add an additional baseline by post-training the previous state-of-the-art model Ctrl-World [1] with its official configuration. We present the results of Ctrl-World below. DreamDojo-2B surpasses Ctrl-World by a large margin across all evaluation sets.
>
> In-lab Eval:
> | Model | PSNR$\uparrow$ | SSIM$\uparrow$ | LPIPS$\downarrow$ |
> | - | - | - | - |
> | Ctrl-World | 20.075 | 0.719 | 0.215 |
> | DreamDojo-2B | 21.114 | 0.774 | 0.222 |
>
> EgoDex Eval:
> | Model | PSNR$\uparrow$ | SSIM$\uparrow$ | LPIPS$\downarrow$ |
> | - | - | - | - |
> | Ctrl-World | 19.637 | 0.735 | 0.205 |
> | DreamDojo-2B | 20.411 | 0.775 | 0.226 |
>
> DreamDojo-HV Eval:
> | Model | PSNR$\uparrow$ | SSIM$\uparrow$ | LPIPS$\downarrow$ |
> | - | - | - | - |
> | Ctrl-World | 17.920 | 0.694 | 0.231 |
> | DreamDojo-2B | 18.813 | 0.747 | 0.238 |
>
> Counterfactual Eval:
> | Model | PSNR$\uparrow$ | SSIM$\uparrow$ | LPIPS$\downarrow$ |
> | - | - | - | - |
> | Ctrl-World | 19.391 | 0.736 | 0.216 |
> | DreamDojo-2B | 20.907 | 0.787 | 0.192 |
>
> **(Q2) Use DreamDojo for world-model-based RL.** We agree that using DreamDojo for model-based reinforcement learning is a promising application, as our results show that DreamDojo can provide reliable evaluations and successfully select better actions at test time. On the other hand, the scope of this paper is mainly focused on how to enable generalization from large-scale human egocentric videos and achieve real-time interactivity. We plan to explore this application in future work, following recent approaches that primarily use in-domain world models for RL [2-5].
>
> **(Q3) Metrics that measure how well the predicted frames are aligned with the input actions.** To address this concern, we conduct human preference evaluations on the first three evaluation sets used in Table 2 in terms of physics correctness and action following. We invite 8 volunteers for this experiment. Please refer to Appendix B Line 865-870 for a more detailed evaluation protocol. We choose Comos-Predict 2.5 as the anchor model to obtain the win rates. The results are shown below.
>
> (metrics in table: *physics correctness* / *action following*)
> | Data Mixture | In-lab Eval | EgoDex Eval | DreamDojo-HV Eval |
> | - | - | - | - |
> | In-lab | 53.125% / 52.5% | 49.875% / 52.8975% | 47.75% / 48.375% |
> | In-lab + EgoDex | 56.5% / 56.5% | 57.24% / 54.5625% | 52% / 55.5% |
> | In-lab + EgoDex + DreamDojo-HV | 59.375% / 59.5% | 56.5% / 51.2875% | 54.25% / 56.75% |
>
> [1] Ctrl-World: A Controllable Generative World Model for Robot Manipulation
>
> [2] WMPO: World Model-based Policy Optimization for Vision-Language-Action Models
>
> [3] WoVR: World Models as Reliable Simulators for Post-Training VLA Policies with RL
>
> [4] VLA-RFT: Vision-Language-Action Reinforcement Fine-tuning with Verified Rewards in World Simulators
>
> [5] World-Env: Leveraging World Model as a Virtual Environment for VLA Post-Training

---

> > ### Author Rebuttal · Reviewer_KAD9 · 2026-04-03
> >
> > The authors have adequately addressed my concerns. The added Ctrl-World baseline and human preference evaluation on action following strengthen the experimental validation. I am raising my score from 4 to 5.

---

> > > ### Author Response · Authors · 2026-04-03
> > >
> > > Thank you for your constructive suggestions and raising the score. We will include all results and discussions in the revision.

---

### Official Review · Reviewer_xnQ9 · 2026-03-11

**Soundness:** 4
**Presentation:** 3
**Significance:** 3
**Originality:** 3
**Overall Recommendation:** 5
**Confidence:** 4

**Summary:**

This paper investigates learning world models via pre-training on large-scale human videos and then post-training on action-conditioned data.
They're trying to see how we can scale pre-training without relying on action-labelled video data, which is costly to obtain.
First, they present the DreamDojo-HV human video data set, which comprises 44k hours of human video.
They show that pre-training on this data set with a self-supervised method for extracting per-frame continuous latent actions leads to improved performance when compared to simple video pre-training (i.e. no actions).
The method resembles the inverse dynamics model (IDM) method, which has become popular for extracting actions from video sequences, except that they formulate it as a VAE with an action prior for regularisation.
Next, they show how to post-train the model on action-labelled data and then they show how to distil the teacher model into a faster autoregressive student using self-forcing.
They also provide a thorough set of experiments.
First, they evaluate the effect of using different data mixtures on predictive performance.
They then compare different methods for including action conditioning during pre-training and then they show that distillation leads to a fast student.
They then show that the world model can be used for both policy evaluation and planning.

**Compliance With Llm Reviewing Policy:**

Affirmed.

**Final Justification:**

I have read the other reviews and the authors’ rebuttal, and I remain convinced that this paper should be accepted.

While the work builds on existing ideas, the particular combination of these methods, the scale at which they are applied, and the release of the dataset -- along with its evaluation in the context of world modelling -- constitute a meaningful and novel contribution.

**Key Questions For Authors:**

1. In the tables, how are these values calculated?
Do the metrics (PSNR/SSIM/LPIPS) represent the values of a single frame (e.g. the last frame), or a statistic aggregating over the trajectory (e.g. mean over the sequence)? This should be clear from the tables' captions.

**Limitations:**

Yes

**Strengths And Weaknesses:**

Overall, I think this paper should be accepted.
First off, it is well written and easy to follow.
Second, it investigates an interesting problem that is of value to the community.
That is, what is the best way to scale world model pre-training given the difficulty of scaling action-labelled data.
This is backed up with an extremely thorough set of experiments covering i) predictive performance (PSNR/LPIPS/SSIM), ii) correlating policy evaluation inside the world model with the real world, and iii) using the world model for model-based planning.
Arguably, all the things we would like to do with the world model.
They are also releasing the data set, which will be of value to the community.

One could argue that the method is incremental in the sense that it mainly takes ideas from existing works and combines them.
However, the release of the data set and its evaluation on world modelling are novel.

One point of concern is that I think some extra citations are needed to give proper attribution.
First of all, in Section 3.3.2, the authors propose to learn the latent action using an inverse dynamics model (IDM), although they never explicitly say it's an IDM.
This section should cite prior works that leverage IDM's for extracting latent actions from videos, e.g. Genie, VPT.
Second, in Section 3.3.4, the authors propose to use the distribution matching loss and generate frames longer than the teacher's horizon was.
I'm fairly certain this is what was done in APT2.
Whilst the authors have already cited this paper, it needs to be cited in this section.

# Minor corrections
1. The text in Figure 2b needs to be made larger.
2. Line 256, the citation should not be in brackets. Use `\citet` instead of `\citep`.
3. Line 242 right col, "marching" should be "matching".

# References
- Baker, Bowen, et al. "Video pretraining (vpt): Learning to act by watching unlabeled online videos." Advances in Neural Information Processing Systems 35 (2022): 24639-24654.
- Lin, Shanchuan, et al. "Autoregressive adversarial post-training for real-time interactive video generation." Advances in Neural Information Processing Systems 38 (2025).
- Bruce, Jake, et al. "Genie: Generative interactive environments." Forty-first International Conference on Machine Learning. 2024.

---

> ### Author Rebuttal · Authors · 2026-03-30
>
> Thanks for the feedback. We address each concern below and will include all results in the revision.
>
> **(W1) The method mainly takes ideas from existing works.** While our work is inspired by several previous works, we are the first to curate such a large-scale dataset and systematically validate all designs at this scale. To the best of our knowledge, we present the first model of its kind that demonstrates exceptional generalization ability and can perform long-horizon rollout at a real-time speed. We will open-source the whole training and evaluation pipeline and hope it will pave the way for general-purpose robot world models.
>
> **(W2) Missing citations.** We will cite Genie, VPT, and APT2 in the appropriate places as suggested.
>
> **(W3) Mirror corrections.** Thank you for pointing those out. We will fix all mentioned typos as suggested.
>
> **(Q1) How the metrics are calculated.** The PSNR, SSIM, and LPIPS metrics are calculated frame-wise and then averaged over all video frames in the evaluation sets. We will clarify this in the revision.

---

> > ### Author Rebuttal · Reviewer_xnQ9 · 2026-04-03
> >
> > I thank the authors for the rebuttal.
> >
> > I continue to support this paper for acceptance.
> >
> > One comment on the metrics (PSNR/LPIPS/SSIM/etc). When evaluating a checkpoint, why don't you sample a set of trajectories and calculate the mean over the samples?

---

> > > ### Author Response · Authors · 2026-04-04
> > >
> > > Thank you for your constructive suggestions and the support for acceptance. We indeed have 100 trajectories in each evaluation set (Line 816-821), and we generate 48 future video frames starting from the ground truth initial frame of each trajectory. All the resulting frames will be used for frame-wise metric calculations and averaged afterward. We hope this addresses your concern. We will further clarify this in the revision and open-source all evaluation codes for reproducibility.

---

### Official Review · Reviewer_8QHB · 2026-03-12

**Soundness:** 4
**Presentation:** 4
**Significance:** 3
**Originality:** 3
**Overall Recommendation:** 5
**Confidence:** 4

**Summary:**

This paper presents a robotic world model that uses latent actions to leverage large-scale unlabeled human videos. In-label embodiment-specific data can later be collected to fine-tune the action projection layer, enabling adaptation to specific robotic embodiments. The resulting model is evaluated in several downstream applications, including live teleoperation, policy evaluation and model-based planning.

**Compliance With Llm Reviewing Policy:**

Affirmed.

**Final Justification:**

My primary concern regarding this paper was the incorporation of failure data for a more faithful evaluation. This issue has been addressed during the rebuttal phase through additional experiments. Therefore, I maintain my original rating and recommend accepting the paper.

**Key Questions For Authors:**

1. What architectural or training choices (e.g., KL scheduling) were used to ensure stable latent action discovery and avoid posterior collapse? If applicable, empirical observations or ablation results would help clarify this point.
2. The dataset appears to only contain successful human demonstrations. Were there attempts to include failure cases (either from human videos or from robot-collected data)? If so, how did such data affect the model's ability to represent unsuccessful interactions or recover from mistakes?
3. During live teleoperation, how long can the model generate coherent rollouts before noticeable degradation occurs?
4. The embodiment adaptation stage uses approximately 55 hours of GR-1 data for post-training. How sensitive is performance to the amount of post-training data? In particular, is this scale of data generally required for new embodiments, or does performance remain stable under smaller adaptation datasets?
5. The paper attributes the performance improvements to transferable interaction knowledge learned from large-scale human videos. However, the current experiments do not clearly disentangle this effect from a simpler alternative explanation: the gains may primarily come from generic large-scale visual pretraining rather than interaction-specific dynamics. A more controlled comparison (e.g., videos with minimal interaction vs. manipulation-heavy videos) would help clarify whether the benefit truly comes from transferable interaction knowledge.

**Limitations:**

yes

**Strengths And Weaknesses:**

Strengths:
1. The paper introduces a large-scale dataset of approximately 44k hours of human manipulation videos. The overall pipeline for leveraging this dataset, including latent action discovery, embodiment adaptation, and inference distillation, is clearly described and logically organized.
2. Extensive empirical evaluation of the trained world model, including video generation quality under out-of-distribution settings, teleoperation and policy evaluation, providing a thorough demonstration of the model's capabilities as a robust simulator.

Weaknesses:
1. The latent action learning relies on an information bottleneck objective implemented via a VAE-style KL regularization. In standard VAEs, the latent representation may capture environmental noise rather than desired latent actions, and is prone to posterior collapse in face of a strong decoder. The manuscript provides limited discussion on how these issue are mitigated, limiting the scalability of this approach when in-the-wild human videos are further included.
2. Video generation quality is primarily evaluated using standard video metrics, which do not fully capture structural and geometric consistency in generated scenes. For example, deformable and inconsistent object shapes (e.g., apples in the accompanying video) can be observed qualitatively.
3. For policy evaluation, three things deserve further consideration. (1) The task selection (described in Appendix B: Policy Evaluation Details) focuses on relatively contained pick-and-place scenarios. The evaluation metric for these tasks is the number of fruits successfully placed, an outcome-based metric that does not capture intermediate aspects of task execution, such as completion time or grasp stability. As a result, the experimental validations do not fully reflect the fidelity of the simulated interaction process, particularly for longer-horizon or more dexterous tasks. (2) DreamDojo tends to produce optimistic outcome estimates despite achieving high Pearson correlation and low MMRV, and authors neither provide analysis nor practical ways to alleviate this issue. The manuscript notes this behavior but provides limited analysis of its causes or potential mitigation strategies. For multi-stage tasks with temporal dependency, this overly optimistic behavior will eventually make the simulator unreliable.

---

> ### Author Rebuttal · Authors · 2026-03-31
>
> Thanks for the feedback. We address each concern below and will include all results in the revision.
>
> **(W1&Q1) How to mitigate capturing environmental noise and posterior collapse.** While reducing environmental noise is important, containing sufficient information about real actions is more important [1]. In other words, since there is a post-training stage, our latent action doesn't have to be clean to serve as a good proxy action, but it’d better to represent most human actions during pretraining. Regarding the second issue, our latent action model is less likely to collapse because: (1) The extremely large scale of our dataset makes it challenging to predict future human videos based solely on the previous frame, as different collectors may have varying behaviors. There is almost no cue indicating the next step of the current person. Thus, our model must learn from latent actions to minimize the prediction loss. (2) The parameter $\beta$ is adjusted to allow sufficient information to pass the bottleneck, ensuring that the latent actions capture meaningful information.
>
> **(W2) Structural and geometric consistency metric.** We supply human evaluations to address this concern. For the physics correctness evaluation, we ask the participants to focus on object permanence, structural and geometric consistency, and contact causality. Please refer to our rebuttal to reviewer KAD9 Q3. The results show that our approach improves the structural and geometric consistency.
>
> **(W3) Policy evaluation experiments can be further enhanced.** (1) In the paper, we choose fruit packing as a representative long-horizon task to demonstrate the initial promise of DreamDojo for reliable policy evaluation. We set a time limit of 2 minutes to penalize policies that complete the task too slowly. (2) DreamDojo is still not perfect in simulating nuanced failures (Line 773-775). Adding policy rollouts to the post-training stage would be a promising solution (Line 775-776, also see our results for Q2). Expanding the evaluation to include more complex task settings is worth exploring, and we leave this for future work.
>
> **(Q2) Effect of training failure cases.** To study the effect of incorporating failure cases into the post-training stage, we post-trained two variants of DreamDojo on the bimanual YAM datasets. One variant was trained on 35 hours of demonstration data, while the other combined 35 hours of demonstration data with 5 hours of policy rollouts, where about 5/6 were failures. Both variants are trained for 20k steps and the mixture ratio of demonstration and rollout is 2:1 for the second variant. We test PSNR of both models on a held-out demonstration set and a failure rollout set. The results demonstrate that including failure cases improves performance in both situations, especially the unsuccessful trajectory simulation.
> | Eval Set | w/o rollout | w/ rollout |
> | - | - | - |
> | Success | 21.522 | 21.547 |
> | Failure   | 20.656 | 22.220 |
>
> **(Q3) How long the live teleoperation can continue before degradation.** The live teleoperation can continue stably for more than 10 seconds, as long as the magnitude of the action inputs remains within a reasonable range.
>
> **(Q4) Effect of the scale of the post-training data.** When we post-trained DreamDojo on 88 hours of GR-1 data and 33 hours of AgiBot fruit packing data, we found that the controllability of the resulting models was quite strong. While this depends on the complexity of the target action space, we believe that tens of hours of post-training data should be sufficient for adaptation in general. We also conducted an ablation using 10% and 50% of the GR-1 data for 15k steps post-training and evaluated the post-trained models (with and without our human video pretraining) on the In-Lab Eval in PSNR. The models with pretraining outperformed those without in all cases. Surprisingly, we found that our pretraining made the model less sensitive to smaller post-training datasets, while Cosmos-Predict2.5 post-trained on 8 hours of GR-1 data may have an overfitting issue.
> | Data Scale | w/o pretrain | w/ pretrain |
> | - | - | - |
> | 8h | 18.889 | 21.261 |
> | 44h | 19.081 | 21.175 |
> | 88h | 19.074 | 21.109 |
>
> **(Q5) Pretraining on generic videos vs. manipulation-heavy videos.** In fact, the main experiments in this paper, Cosmos-Predict 2.5 vs. Ours (Cosmos-Predict 2.5 + our human video pretraining), already shows that interaction-specific video pretraining can benefit world modeling for robot manipulation. Cosmos-Predict 2.5 is pretrained on generic Internet videos, where manipulation skills are not the primary focus. In contrast, our method resumes from this pretraining using egocentric interaction videos, thereby validating that manipulation-heavy videos help the transfer of interaction knowledge. We will continue exploring this key factor in future work.
>
> [1] Latent Action Learning Requires Supervision in the Presence of Distractors

---

> > ### Author Rebuttal · Reviewer_8QHB · 2026-04-01
> >
> > I thank the authors for adding the additional experiments and for their efforts in addressing my concerns. I will maintain my positive rating of this paper.

---

> > > ### Author Response · Authors · 2026-04-02
> > >
> > > Thank you for your constructive suggestions. We will include all results and discussions in the revision.

---

### Official Review · Reviewer_57Nw · 2026-03-13

**Soundness:** 2
**Presentation:** 3
**Significance:** 1
**Originality:** 1
**Overall Recommendation:** 3
**Confidence:** 4

**Summary:**

The authors tackle the problem of learning video world models with applications to contact-rich robotics tasks. They address this problem by designing a foundation model for dexterous manipulation tasks, including unseen instances. This model is trained on 44k hours of human ego-centric videos that capture a wider range of categories. The authors propose to build on an existing foundation model with . Their experiments test the model's accuracy against the existing foundation model in out-of-distribution cases. They also evaluate, focussing on AgiBot fruit picking, their proposed model on teleoperation, planning, and policy evaluation.

**Compliance With Llm Reviewing Policy:**

Affirmed.

**Final Justification:**

While the additional results add more context, the following concerns remain around model quality and downstream benefits:

The stated confidence intervals for Table 2 do not convincingly support the authors' claim about significant OOD improvement with the suggested scaling of the data mixture.

It remains unclear how the human evaluation results add support to this claim. If my understanding is correct, the results show win-rates that suggest that humans have a high degree of uncertainty (choosing almost with random chance).

The claims about downstream performance remain weakly supported due to a) narrow evaluation b) limited baseline evaluation. For example, the authors claim that DreamDojo can serve as a reliable real-world simulator for policy improvement. However, Figure 4 suggests that achieving high success in DreamDojo correlates with relatively weak success rates in the real world (100% simulated success => 50% real-world success). An evaluation including other world model baselines (e.g., Ctrl-World on downstream tasks) and more tasks could reveal significance if there is improvement over the state-of-the-art in downstream tasks, assuming a gap exists.

**Key Questions For Authors:**

1. The margins of model performance appear to quite small; e.g., around 2-5 tenths in Table 2. What are the confidences of these numbers? Does this constitute a large improvement?
3. Table 1 compares the number of skills and scenes across different datasets. For the larger datasets (including DreamDojo), what definition of skill is used? What about scene? How do these definitions affect the calculations in the table?
4. What is the relative performance of pre-training versus post-training? How far does the pre-training carry downstream performance?
5. Table 5 shows human preference evaluation. How is this evaluation relevant to the problem posed in the introduction?

**Limitations:**

Yes

**Strengths And Weaknesses:**

Strengths
1. The authors present a large-scale dataset with tens of thousands of hours of human videos which has potential to benefit foundation-model-based robot learning. The dataset appears to be composed of many categories of tasks and scenes in human-centric environments.
2. The organisation of the paper and layout of figures and tables supports readability.
3. The evaluation of the model's accuracy is extensive; including multiple error and performance metrics.

Suggested improvements

*tl;dr  The current evaluation focusses on model predictive performance, which while important, does not directly imply guarantees for policy improvement [1].* The authors should consider expanding their evaluation in order to demonstrate significance of their proposed contributions.

1. Downstream benefits are unclear from the evaluation results in Figures 4-5. What are the models listed along the x-axis? The reader cannot meaningfully orient the performance of DreamDojo otherwise.
2. Trade-off between latency and overall performance (ln 400) should be more clearly stated. What is this tradeoff and how do you justify having additional latency compared to other approaches?
3. A more specific question (Q5 in Sec 4.1) could help direct the evaluation towards a more meaningful result. For example, "Does DreamDojo improve task success compared to multiple baselines [2] in X & Y domain?", where a domain comprises multiple tasks.
4. Figure 2 shows the composition of the proposed dataset, but could benefit from significant editing. The plots/histograms are difficult to read. Additionally, the notion of skill seems inconsistent with the literature where, for example, pick-and-place would be considered a skill comprised of pick action, move action, and place action.
5. Table 1 compares the size of different datasets, but citations are missing which would add to clarity. Some of these datasets e.g., Language-Table are missing references. Also, some of the selected datasets (Nymeria) appear tangential to manipulation domain.
6. Table 4 caption (~ln357) should be reworded; e.g., "Our distilled model is capable of real-time inference (10.93 FPS)".
7. Table 5 shows marginal improvements. Based on the absolute performance, one percent appears an insignificant improvement on human preference. Furthermore, there are no confidence intervals.
8. When referencing the appendix, please cross-reference the appropriate sections.
9. The authors claim that DreamDojo-HV is the largest ego-centric human video dataset, which is quite impressive. Size is one thing, but it is not exactly clear what about the composition of this dataset makes it beneficial for learning (skill transfer in particular) [1]. The authors add that they improve on past datasets with increased diversity - in number of skills and scenes. However, this appears to be simply scaling up on the categorical level, perhaps by changing the definition of "skill" (Figure 1 and 5 where the count of skills is a GPT-estimate).

[1] What Matters in Learning from Large-Scale Datasets for Robot Manipulation, Saxena et al. 2025 ICLR

---

> ### Author Rebuttal · Authors · 2026-03-31
>
> Thanks for the feedback. We answer each concern below and will include all results in the revision.
>
> **(W1) Unclear results in Fig 4-5.** These two experiments prove that DreamDojo can give reliable feedback for policy evaluation and planning. The models along the x-axis are different policy checkpoints from one training (Line 390). They have varying success rates but each of them may excel at certain states. At inference, we ensemble the action proposals from these checkpoints and use DreamDojo to select the most promising one to execute. This results in a 5.33% improvement over the best policy without additional training.
>
> **(W2) Latency trade-off.** Running the policy ensemble, world model, and reward model indeed introduces extra latency compared to typical policy inference. However, this notably improves success rate *without additional training*. Although the extra latency is non-negligible at present, this searching is not required at every timestep [1], and several techniques can be applied to boost the speed [2]. We will explore these in future work.
>
> **(W3) Make Q5 in Sec 4 more specific.** We will specify it as “Can DreamDojo empower downstream applications, including policy evaluation, test-time search, and live teleoperation?”
>
> **(W4) Improve Fig 2.** We will make the plots more intuitive. We indeed decompose pick-and-place into atomic skills as Fig 2b[Right] shows. We will clarify this in the revision.
>
> **(W5) Missing citations in Table 1.** We do have the papers cited but hidden them in the table for formatting. We will re-add them. We include Nymeria as a representative of existing large-scale human datasets that cover manipulation activities.
>
> **(W6) Reword Table 4 caption.** We will update it accordingly.
>
> **(W7) Table 5 improvements.** Thanks for pointing out this potential misunderstanding, we will clarify Table 5. As Line 823 describes, we evaluate models on two metrics, physics correctness (row 1) and action following (row 2). Each column describes a pairwise comparison; eg, compared to Cosmos-Predict2.5, DreamDojo-2B achieves win rates of 62.5% in physics correctness and 63.45% in action following (much higher than one percent). Confidence intervals:
> | | | |
> | - | - | - |
> | 62.50% ($\pm$0.15%) | 73.50% ($\pm$0.09%) | 72.50% ($\pm$0.08%) |
> | 63.45% ($\pm$0.16%) | 72.55% ($\pm$0.15%) | 65.53% ($\pm$0.14%) |
>
> **(W8) Appendix ​​cross-reference.** We will cross-reference the corresponding sections.
>
> **(W9) What makes the dataset beneficial.** Table 2 shows that gradually increasing the diversity (EgoDex adds objects and skills, and HV increases the diversity in all axes) continuously improves the quality. How to scale the data more effectively requires further study. See Q2 answer for our skill definition.
>
> **(Q1) Table 2 improvements.** The confidence intervals are around $\pm$0.4, 0.013, and 0.008 for PSNR, SSIM, and LPIPS, respectively. We will provide the complete intervals for all results in the revision. To address the concern of improvement, we perform human evaluations on the same evaluation sets. Please refer to our rebuttal to reviewer KAD9 Q3. The results show that expanding human data lead to non-trivial improvements in physics and controllability.
>
> **(Q2) Table 1 definitions.** Our definition of skill aligns with AgiBot-World [3] (see our Fig 2b[Right] and their Fig 3c). We define each new environment (eg, a new room) as a unique scene, which is also aligned with prior work. As each video is shot in a different place, the total number of scenes equals the number of videos. While GPT estimation is perfectly accurate, we have instructed it carefully and manually checked its reasoning. We will add more details in the revision.
>
> **(Q3) Relative performance of pretraining.** Table 2 (Cosmos-Predict2.5 vs. DreamDojo) and Table 3 (w/o pretrain vs. latent action) show that our pretraining consistently improves simulation quality. To verify if its benefit can propagate to downstream, we distill a model post-trained from Cosmos-Predict2.5 on AgiBot fruit packing. We then evaluate its usefulness together with our distilled DreamDojo-2B following the model-based planning setup in Sec 4.5. The success rates below show that both models help, but the model with pretraining is better at selecting the correct actions.
> | unified sampling | w/o pretrain | w/ pretrain |
> | - | - | - |
> | 37% | 43% | 70% |
>
> **(Q4) What does Table 5 imply.** The results show that the proposed human video pretraining with latent actions brings notable gains in environment generalization on our out-of-distribution evaluation sets generated by Nano Banana (Line 305-310). As the ground truth videos are unavailable, we use human evaluation for this test.
>
> [1] Seeing Farther and Smarter: Value-Guided Multi-Path Reflection for VLM Policy Optimization
>
> [2] RELIC: Interactive Video World Model with Long-Horizon Memory
>
> [3] AgiBot World Colosseo: A Large-scale Manipulation Platform for Scalable and Intelligent Embodied Systems

---

> > ### Author Rebuttal · Reviewer_57Nw · 2026-04-02
> >
> > The authors addressed all points in my review and related remarks from other reviewers. Given their clarifications about the basis on which skills are defined and interpretation of the human evaluation, I have increased my score. However, the evaluation and its results are not sufficiently convincing of the claims about accuracy and generalisation. The improvements seem marginal under the authors' proposed pre-training recipe. The main contribution seems to be the curation of a large dataset for foundation model pre-training.

---

> > > ### Author Response · Authors · 2026-04-02
> > >
> > > Thank you for your feedback! We are glad to hear that all your concerns have been adequately addressed, and we appreciate your acknowledgment of our contributions in large-scale dataset curation for foundation model pre-training. We will clarify all the aforementioned definitions and include all results in the revision.
> > >
> > > Regarding the insufficient evaluation of accuracy and generalization, could you please specify what particular experiments would be more convincing, in addition to the 27% downstream performance jump (43% to 70%) we see in Q3? As part of this, what improvement margins (seen in previous literature) would you consider sufficient? In addition, please note that as a pioneering effort, we have conducted systematic evaluations in out-of-distribution settings with unseen objects and scenarios, and the proposed approach consistently outperforms the baseline. We are happy to add more evidence and greatly appreciate your help in improving our paper. Thank you!
> > >
> > > **After reading the final justification, we would like to add two additional clarifications:**
> > >
> > > - The reviewer may misinterpret the significance of our improvements in human evaluation. Our win rates, obtained from 12 volunteers on 100 samples, were obtained through rigorous protocols to minimize any potential bias (Line 823). Achieving approximately 63% win rates represents a non-trivial improvement, and the $\pm$0.15% confidence interval indicates that the results will consistently fall around 63% if we redo the evaluation. This is by no means a "choosing almost with random chance". The improvement is also clearly demonstrated by our qualitative results in Fig. 9.
> > >
> > > - Regarding the policy evaluation experiment in DreamDojo, we want to highlight that DreamDojo demonstrates a strong correlation with the real world (Pearson r=0.995, MMRV=0.003) and reliably judges the goodness of different checkpoints. Due to the limit of our post-training data distribution, DreamDojo cannot perfectly replicate absolute success rates in the real world, which is a common limitation in existing world models [4,5]. Unlike these works, we have discussed the potential solutions (Line 775) and provided preliminary results in our rebuttal to reviewer 8QHB Q2. However, this does not diminish the fact that DreamDojo can accurately rank policies and is valuable for policy evaluation.
> > >
> > > [4] Evaluating Gemini Robotics Policies in a Veo World Simulator
> > >
> > > [5] WorldEval: World Model as Real-World Robot Policies Evaluator

---

### Decision · Program_Chairs · 2026-04-30

**Decision:**

Accept (spotlight)

**Comment:**

The paper received mostly positive reviews. Reviewers appreciated the scale and potential value of the DreamDojo-HV dataset, as well as the overall engineering of the pipeline for leveraging large-scale unlabeled human videos for robot world modeling. The experimental section was generally viewed as broad and practically relevant, covering predictive quality, teleoperation, policy evaluation, and planning, and the rebuttal further strengthened the paper by clarifying evaluation details, adding stronger baselines, and providing additional evidence on action following and failure cases.

At the same time, the main contribution is best understood as a large-scale data and systems contribution rather than a fundamentally new world-modeling method, since several core components build on existing ideas. Some concerns also remain about the strength of the downstream claims, particularly around simulator fidelity and policy improvement. Overall, however, I believe the paper makes a meaningful contribution and merits acceptance.